

# Exploring the jamming transition
# over a wide range of critical densities

**Misaki Ozawa, Ludovic Berthier and Daniele Coslovich**

L2C, Université de Montpellier, CNRS, Montpellier 34095, France

## Abstract

We numerically study the jamming transition of frictionless polydisperse spheres in three dimensions. We use an efficient thermalisation algorithm for the equilibrium hard sphere fluid and generate amorphous jammed packings over a range of critical jamming densities that is about three times broader than in previous studies. This allows us to reexamine a wide range of structural properties characterizing the jamming transition. Both isostaticity and the critical behavior of the pair correlation function hold over the entire range of jamming densities. At intermediate length scales, we find a weak, smooth increase of bond orientational order. By contrast, distorted icosahedral structures grow rapidly with increasing the volume fraction in both fluid and jammed states. Surprisingly, at large scale we observe that denser jammed states show stronger deviations from hyperuniformity, suggesting that the enhanced amorphous ordering inherited from the equilibrium fluid competes with, rather than enhances, hyperuniformity. Finally, finite size fluctuations of the critical jamming density are considerably suppressed in the denser jammed states, indicating an important change in the topography of the potential energy landscape. By considerably stretching the amplitude of the critical "J-line", our work disentangles physical properties at the contact scale that are associated with jamming criticality, from those occurring at larger length scales, which have a different nature.


## 1  Introduction

Granular materials such as grains, pinballs, and large colloids flow when some external forces are applied. However, when the volume fraction of these particles is increased above a certain value, particle motion is no longer allowed and these systems become amorphous solids. This mechanical transition corresponds to a jamming transition because it occurs in the absence of thermal fluctuations. The critical volume fraction of this critical "J-point" is $\phi_J$, and the associated critical properties have been intensively studied in the last few decades [1–4]. In particular, structural properties of jammed states have been examined over a wide range of length scales from microscopic to macroscopic, providing a starting point for understanding the mechanical and rheological properties of jammed systems [3,5–8].

In practice, the structural properties of jammed systems can be sorted out by the typical length scale or wave number $k$ that is being probed. First, the *contact scale*, corresponding to $\delta \to 0$, where $\delta$ is the typical gap between particles for $\phi < \phi_J$, or to the typical overlap between particles for $\phi > \phi_J$. The corresponding wave number is $k \sim 1/\delta \to \infty$. A remarkable property at the contact scale is isostaticity, which implies that the average number of contacts per particle, $Z$, becomes twice the number of spatial dimensions exactly at $\phi = \phi_J$. In addition, a critical power law behavior characterizes the pair distribution $g(r)$ near contact at $\phi_J$ [3,5,9,10]. The isostatic nature of the system is at the core of theoretical descriptions of the jamming transition [3,4,11]. Second, one can consider distances at the *neighbor scale*, corresponding to $k \sim 2\pi/\overline{\sigma}$, where $\overline{\sigma}$ is the averaged particle diameter. The major goal at this scale is to properly characterize the local geometry of amorphous configurations [12–14], to possibly define appropriate order parameters for the jamming transition [15,16]. The role of local crystalline and icosahedral order in monodisperse packings has been discussed before [2,17–21]. In particular, suitable modifications of bond orientational order parameters have revealed local crystalline order in (non-isostatic) jammed packings [18,19], while icosahedral order is of more limited extent [20,21]. Finally, we can analyse the *large scale* corresponding to $k \to 0$, where fluctuations over the entire sample are considered. It has been reported that systems prepared at the jamming transition have unexpected density fluctuations at large scales, possibly corresponding to hyperuniform behavior. Physically, hyperuniformity implies that the amplitude of volume fraction fluctuations at wave number $k$ vanishes as $k \to 0$ [22,23]. However, several recent works have reported numerical evidence that hyperuniformity is not strictly obeyed at the jamming transition [24–26].

The structural properties mentioned above have been studied at "the" jamming transition.

However, theoretical studies have established that $\phi_J$ is not uniquely determined but is in fact strongly protocol dependent [15,27–41]. This implies that the jamming critical point, J-point, actually corresponds to a line of critical points, thus forming a "J-line" [31]. A simple way of exploring the J-line is to perform compressions of hard sphere configurations using different compression rates. A more controlled method consists in first thermalising the hard sphere fluid at finite temperature at some volume fraction and then performing a rapid compression towards jamming [31, 36, 42]. By varying the volume fraction of the parent fluid, a finite range of jamming volume fractions can be explored, while easily keeping crystallisation under control.

Although the existence of a J-line is well accepted, much less is known about how structural properties of jammed systems may change along the J-line. Some previous studies reported that the structural properties at the contact and neighbor scales hardly change [30,31,33,43] and thus one might conclude that the structural properties are essentially invariant along the J-line. Scaling theories of the jamming transition are based on this assumption [3, 4, 11]. On the contrary, some other studies pointed out that there are tiny but systematic structural changes [34, 36], in particular at the neighbor scale. To our knowledge, the evolution of the large scale structural properties along the J-line has not been studied in detail. Analyzing the evolution of structure along the J-line is a numerical challenge, because changing the value of $\phi_J$ requires changing the typical preparation timescale over orders of magnitude, and even large numerical efforts may yield relatively minute changes to the value of $\phi_J$. Therefore, previous studies have accessed finite, but quantitatively modest, variations of $\phi_J$ along the J-line.

In this paper, we develop a numerical strategy that allows us to stretch the extension of the J-line of frictionless hard spheres considerably. As a result, we can characterize the variation of structural properties with $\phi_J$ over an unprecedentedly broad range of volume fractions, $\phi_J \approx 0.65 - 0.70$. This range is at least 3 times wider than in any previous study of frictionless jammed packings [31, 40]. The decisive factor allowing the present analysis is the recent development of an efficient thermalisation algorithm for polydisperse hard sphere fluids. We have recently shown that this approach allows the equilibration of very dense fluid states [44], bypassing any alternative method by many orders of magnitude. Here, we use these deeply thermalised fluid configurations as starting point for rapid compressions towards jammed states.

We find that at the contact scale, both isostaticity and the same critical behavior of the radial distribution function hold over the entire J-line, thus corroborating previous results [31, 33]. To characterize the structure at the neighbor scale, we apply analysis tools appropriate to polydisperse packings, based on the detection of locally favored structures [45, 46]. Our key finding is the detection of distorted local icosahedral structures that become increasingly numerous as $\phi_J$ is increased, so that about 80 % of the particles form such structures in our densest packings. This suggests that this local structural motif might be a key geometric factor allowing the production of very dense jammed packings [20]. At the large scale, we find that a nearly hyperuniform behavior [47] for the smallest $\phi_J$, but deviations from hyperuniformity increase rapidly as $\phi_J$ is increased, suggesting that optimized packings display stronger volume fraction fluctuations at large scale. On the other hand, finite size effects of the mean value of $\phi_J$ are considerably suppressed when $\phi_J$ is large. Overall, our work disentangles contact scale properties that are deeply connected to the criticality of the jamming transition, to structural properties at larger scale which evolve significantly along the J-line, and have therefore a distinct physical origin.

This paper is organized as follows. In Sec. 2, we describe the details of the simulation methods. The resulting extended range of jamming densities is discussed in Sec. 3. We discuss the physics at contact scale (isostaticity, pair correlation) in Sec. 4, the physics at the local scale

(bond orientational order, locally favored structures, rattlers) in Sec. 5, and at the large scale (hyperuniformity, global finite-size effects) in Sec. 6. We conclude our paper and discuss our results in Sec. 7.

## 2 Numerical methods

### 2.1 The model

We employ the standard model of additive frictionless hard spheres in three dimensions [48]. The pair interaction is zero for non-overlapping particles, infinite otherwise. We use a continuous size polydispersity, where the particle diameter $\sigma$ is distributed according to $f(\sigma) = A\sigma^{-3}$, $\sigma \in [\sigma_{\min}, \sigma_{\max}]$, where $A$ is a normalization constant. Following previous work [44], we use the size polydispersity $\Delta = \sqrt{\overline{\sigma^2} - \overline{\sigma}^2}/\overline{\sigma} = 23\%$, corresponding to $\sigma_{\min}/\sigma_{\max} = 0.4492$, where $\overline{\cdots} = \int d\sigma f(\sigma)(\cdots)$. We use $\overline{\sigma}$ as the unit of length. $f(\sigma)$ used in this study is shown in Fig. 6(a). We simulate systems composed of $N$ particles in a cubic cell with periodic boundary conditions. We mainly use the system sizes $N = 1000$ and $8000$, but we also perform selected simulations for $N = 150, 300, 600, 2000, 4000$ to systematically investigate the finite-size effects described in Sec. 6. The state of the hard sphere system is uniquely characterized by the volume fraction $\phi = \pi N \overline{\sigma^3}/(6V)$, where $V$ is the volume of the system. For the fluid state, we measure the reduced pressure $p = P/(\rho k_B T)$, where $\rho = N/V$, $k_B$ and $T$ are the number density, Boltzmann constant and temperature, respectively. We set $k_B$ and $T$ to unity. The pressure $P$ is calculated from the contact value of the pair correlation function properly scaled for a polydisperse system [49]. The fluid state has a finite $p$, whereas the jammed state corresponds to $p \to \infty$.

Note that the functional form of $f(\sigma)$ with $\Delta = 23\%$ is chosen so that the system is fairly robust against crystallization and fractionation at extremely high densities [50]. With smaller $\Delta$, the system easily crystallizes when using the efficient swap Monte Carlo method described below. On the other hand, as demonstrated in previous studies, large values of $\Delta$ or different forms of $f(\sigma)$ lead to fractionation at sufficiently high density [51,52]. By contrast, the model parameters employed in this work are optimized to avoid such instabilities and thus enable us to explore the J-line over an unprecedented range of packing fractions.

### 2.2 Equilibration of very dense fluid states

To obtain equilibrium fluid configurations, we perform Monte Carlo (MC) simulations which combine traditional translational particle displacements and non-local particle swaps [50,53–59]. Translational displacements are drawn from a cube of linear side 0.115, and a trial displacement is accepted if it does not create an overlap between particles [48]. For a trial swap move, we randomly pick a pair $(i, j)$ of particles with $|\sigma_i - \sigma_j| < a$ (we choose $a = 0.2$) and attempt to exchange their diameters [50]. The swap is accepted if it does not create an overlap. We perform translational moves with probability 0.8, and swap moves with probability 0.2 [50]. We have previously established [44] that this swap Monte Carlo setting is extremely efficient to thermalise hard sphere fluid states up to very large volume fractions, $\phi_{\text{fluid}} \approx 0.655$. We have checked that all our equilibrium configurations are taken in the fluid state, carefully monitoring possible signs of demixing or crystallization using the same structural tools described below to also characterize jammed states. Therefore, the parent fluid configurations used to produce jammed states all belong to the (metastable) equilibrium fluid branch, and we do not artificially vary the jamming density by introducing partially crystallized or demixed states. Instead, we explore the equilibrium fluid branch over a broad range of densities.

## 2.3 Compression towards the jamming transition

Having prepared equilibrium fluid configurations of hard spheres, we use the non-equilibrium compression algorithm introduced in Refs. [60, 61] to reach the jamming transition. Briefly, this algorithm replaces the hard sphere potential with a soft repulsive harmonic potential, given by

$$v_{ij}(r_{ij}) = \frac{\epsilon}{2}\left[1 - \left(r_{ij}/\sigma_{ij}\right)\right]^2 \theta(1 - r_{ij}/\sigma_{ij}), \tag{1}$$

where $\epsilon$ is the energy scale, $r_{ij}$ is the distance between the spheres $i$ and $j$, $\sigma_{ij} = (\sigma_i + \sigma_j)/2$ is the distance of the two spheres at contact, and $\theta(x)$ is the Heaviside step function. We use $\epsilon$ as the unit of energy. The idea of the algorithm is to alternate instantaneous compression steps and energy minimization to iteratively converge to the jamming density.

During a compression step, we increase the volume fraction of the system $\phi$ by $\delta\phi = 5 \times 10^{-4}$. This compression introduces finite overlaps between particles, such that the potential energy $U = \sum_{i<j} v_{ij}(r_{ij})$ becomes finite. These overlaps are then eliminated by performing an energy minimization. To this end, we apply the conjugate gradient minimization method [62] for $U$.

If the system still has a finite $U$ after minimization, we decrease $\delta\phi$ by factor of 2 and use a series of decompression and compression steps until the overlaps are eliminated. This process is interpreted as pulling the system from trivial local potential energy minima. We stop the algorithm when $\delta\phi < 1.0 \times 10^{-6}$, and the resulting system is essentially a hard sphere jammed packing. However, we find that the algorithm explained so far [61] produces a fraction of slightly hypostatic packings. Thus, we perform an additional process introduced in Ref. [31] to get more accurate isostatic jammed packings. We simulate sequential compression and minimization with $\delta\phi = 1.0 \times 10^{-5}$ until $U/N > 1.0 \times 10^{-6}$. Then, we decompress the system with $\delta\phi = 1.0 \times 10^{-6}$ until $U/N < 10^{-16}$, which determine the jamming transition point [31].

Note that the jamming algorithm employed in this work is based on purely isotropic compressions and decompressions. Thus, stability against shear deformation is not guaranteed [63], but we expect that further optimization against shear would not change our conclusions.

# 3 Extending the range of jamming densities

We present the equation of state of the system in Fig. 1(a). Equilibrium fluid configurations at finite pressure $p$ are compressed using the non-equilibrium algorithm described in Sec. 2.3 towards jammed states at $p \to \infty$. Figure 1(a) demonstrates that the higher the volume fraction of the parent fluid, $\phi_{\text{fluid}}$, the higher the jamming transition volume fraction, $\phi_{\text{J}}$. Thus, enhanced thermalization is the key to extend the J-line.

In Fig. 1(b), we show $\phi_{\text{J}}$ as a function of $\phi_{\text{fluid}}$, varying $\phi_{\text{fluid}}$ over a very broad range. Below $\phi_{\text{fluid}} \lesssim 0.53$, the observed $\phi_{\text{J}}$ is almost independent of $\phi_{\text{fluid}}$, as suggested by our horizontal dashed line. Also, we show the result obtained for a compression starting from a Poisson distributed system of harmonic soft spheres with $\phi_{\text{fluid}} = 0.3$. We find that $\phi_{\text{J}}$ from the Poisson distribution takes the same value as from dilute hard sphere fluid configurations, $\phi_{\text{J}} \simeq 0.655$. Thus, we confirm that the protocol dependence of $\phi_{\text{J}}$ is essentially absent at $\phi_{\text{fluid}} \lesssim 0.53$. This value for $\phi_{\text{J}}$ is consistent with recent numerical results for a similar continuously polydisperse system [65].

As expected, $\phi_{\text{J}}$ starts to depart from the plateau value $\phi_{\text{J}} \simeq 0.655$ when $\phi_{\text{fluid}} \gtrsim 0.53$, and it then monotonically increases with increasing $\phi_{\text{fluid}}$. The largest value we obtain is $\phi_{\text{J}} \simeq 0.7$. A qualitatively similar behavior was observed before in a binary hard sphere mixture [31, 36].

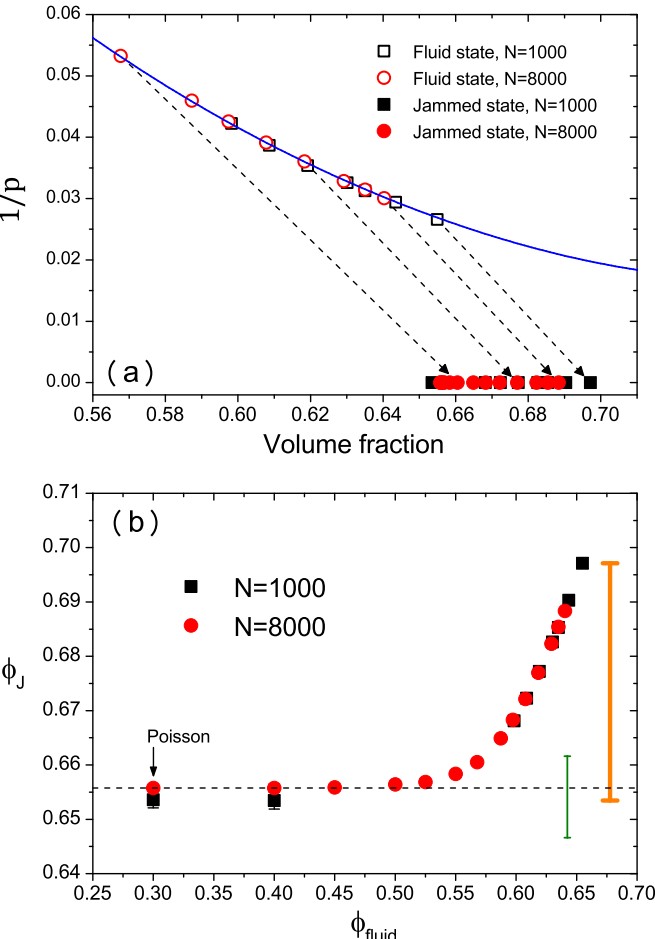

Figure 1: (a) Equation of state of the system for two different system sizes. Equilibrium fluid states are shown as empty points. A modified version of the Carnahan-Starling equation of state for polydisperse systems is drawn as a solid line [64]. Jammed states (filled points) obtained by the compression algorithm are located at $p \to \infty$. The dashed arrows connect an equilibrium parent fluid state to the corresponding jammed state. (b) Evolution of the averaged jamming transition volume fraction $\phi_J$ obtained by the compression from an equilibrium parent fluid with volume fraction $\phi_{fluid}$. We also report $\phi_J$ starting from a Poisson distributed configuration with $\phi_{fluid} = 0.3$. The horizontal dashed line is $\phi_J$ generated from a Poisson distribution for $N = 8000$. The vertical bars correspond to the width of the J-line for this work (thick bar) which is about 3 times larger than previous works using a binary mixture [31] (thin bar).

The variation of the jamming density with the initial fluid density is reminiscent of the variation of the energy of inherent structures with initial temperature for glass-forming materials with a continuous pair interaction [36,66,67]. Note that the finite size effect of $\phi_J$ between $N = 1000$ and 8000 is small at low $\phi_{fluid}$, and it essentially vanishes in the large $\phi_{fluid}$ regime, at least in this graphical representation. The finite size effect of $\phi_J$ will be more systematically examined in Sec. 6. We draw as vertical bars the widths of the J-line obtained in the present study (thick bar) and in a previous study of a binary mixture [31] (thin bar). The extension of the J-line achieved in this work is about 3 times larger than previous works [31, 40]. Thus, we have indeed succeeded in stretching the J-line considerably. In the next sections, we analyze the structural properties of the obtained packings along this stretched J-line.

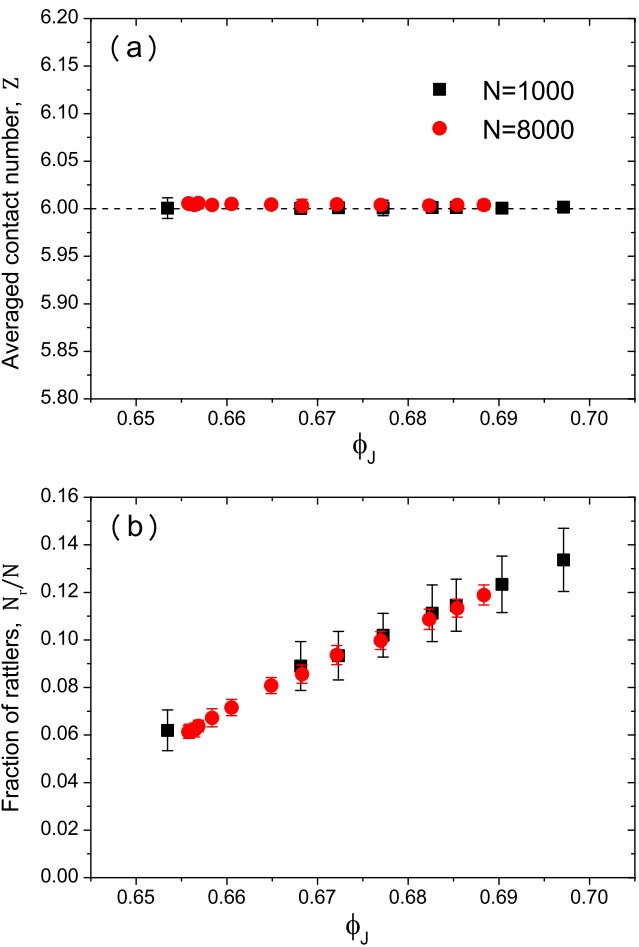

Figure 2: (a): The averaged contact number $Z$ as a function of $\phi_J$ remains close to the isostatic value $Z = 6$ along the entire J-line. (b): The fraction of rattlers, $N_r/N$, increases substantially as a function of $\phi_J$.

## 4 Structure at the contact scale: Isostaticity and pair correlation

### 4.1 Isostaticity

We first examine structural properties at the contact scale. The relevant length scale for $\phi \leq \phi_J$ is the gap between particles, $x - 1$, where $x = r_{ij}/\sigma_{ij}$. A central quantity describing jammed states at this contact scale is the averaged contact number, $Z$. In practice, two particles, $i$ and $j$, are considered in contact if $r_{ij} \leq (1 + a)\sigma_{ij}$, where $a = 1 \times 10^{-5}$ and $5 \times 10^{-6}$ for $N = 1000$ and 8000, respectively [33]. With this definition, $Z$ is given by $Z = N_c/(N - N_r)$, where $N_r$ is the number of the rattlers, $N_c$ is the number of contact pairs among $N - N_r$ particles which make a contact network. We define a rattler as a particle whose contact number is smaller than $d + 1$, where $d$ is the spatial dimension. Note that a determination of rattlers has to be performed iteratively for a given configuration until all rattlers are removed. The isostatic packing is defined by the condition $Z = 2d$ (thus, $Z = 6$ in three dimensions), reflecting the fact that the system is mechanically marginally stable [68].

In Fig. 2(a), we show $Z$ as a function of $\phi_J$ over the entire range of jamming densities that we were able to explore. We find that the isostatic condition, $Z = 6$, indeed holds over the entire J-line obtained by our protocols. In Fig. 2(b) we show the fraction of rattlers, $N_r/N$. It is around 6% at the lowest jamming density, $\phi_J \simeq 0.655$, which is comparable with previous

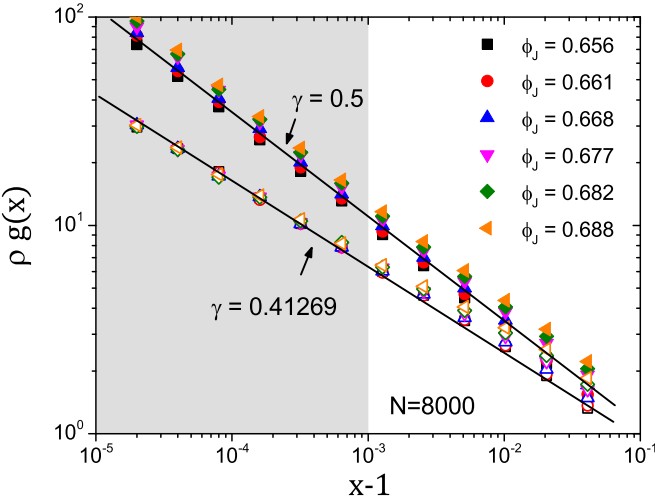

Figure 3: Radial distribution function $\rho g(x)$ near contact, where $\rho = N/V$ and $x = r_{ij}/\sigma_{ij}$. The closed and open symbols are obtained with and without ratters, respectively. The black straight lines characterize a power law behavior, $g(x) \propto (x-1)^{-\gamma}$, with $\gamma = 0.5$ (the upper line) and $\gamma = 0.41269$ (the lower line), respectively. When rattlers are removed, a universal power law with the value predicted by the mean-field theory [70] is observed for all $\phi_J$ values. The gray shaded region marks the regime, $x - 1 \lesssim 10^{-3}$, where critical scaling with $\gamma = 0.41269$ holds.

results using energy minimization in three dimensions for binary [5,31] and polydisperse [69] systems. Interestingly, the fraction of rattlers increases steadily with $\phi_J$. Note that a slight increase of the fraction of rattlers along the J-line was also reported previously in a binary mixture [31,43]. This growth of the number of rattlers might appear counterintuitive at first sight, because rattlers tend to occupy a larger volume [43] and increasing their number should decrease the efficiency of the packing, in contradiction with the results shown in Fig. 2(b). The issue of the increase of the number of rattlers will be further discussed in Sec. 5.3.

## 4.2 Pair correlation function at contact

It is known that the radial distribution function shows a power law critical behavior near contact, $g(x) \propto (x-1)^{-\gamma}$, for isostatic packings. We show $\rho g(x)$, where $\rho = N/V$, as a function of the gap, $x-1$, in Fig. 3 for several $\phi_J$ values along the J-line. Note that we multiply $g(x)$ by $\rho$ to remove the trivial effect of the density change. The data for all $\phi_J$'s follow a power law with an exponent $\gamma = 0.5$, when we compute $g(x)$ from all particles in a given configuration, in particular including rattlers [71]. Instead, when the rattlers are removed and only the particles participating in the contact network are taken into account [72], the $g(x)$ for all $\phi_J$'s now follow a distinct power law which is compatible with the value $\gamma = 0.41269$ predicted by the mean-field theory of the jamming transition [70]. Thus, a careful treatment of the rattlers is essential to assess the critical behavior of the pair correlation function at contact [10,72].

From these observations, we are able to confirm that the critical behavior in $g(x)$ holds over the entire J-line and is therefore universal. This is not a trivial observation. For instance, it may have happened that this property only holds at the lowest end of the J-line, which is the only point where jamming criticality and mean-field predictions had been analyzed before [73]. Notice also that establishing this result was not easy. We found appreciable deviations from a power law behavior for smaller systems at low values of the argument $x - 1$ (not shown). On

the other hand, we also find that by increasing $\phi_J$ the power law regime is entered for lower values of $x - 1$, see Fig. 3. Therefore, observing a power law for large $\phi_J$ is difficult, as the critical regime becomes less easily observed for a given system size as $\phi_J$ is increased.

We expect that other critical behaviors such as a power law distribution of the contact forces are also observed over the entire J-line since these critical behaviors are all directly connected to one another [68,70]. However, this direction is beyond the scope of this paper, since it would require a more important computational effort to prepare the packings exactly at the jamming transition [73].

# 5  Structure at the local scale

Our results so far confirm that the combined use of the swap Monte Carlo technique and polydisperse spheres stretch substantially the J-line, while maintaining isostaticity, $Z = 6$. This already indicates that the system does not have crystalline order, which would produce $Z > 6$ (hyperstatic packings), as explicitly demonstrated in Ref. [33]. For monodisperse spheres, packing fractions comparable and even larger to the ones observed here can be attained. However, these packings are not isostatic and there is evidence that they accumulate local crystalline order beyond a threshold density [18]. Thus, the next question is: How do jammed configurations of polydisperse spheres attain such large densities while remaining amorphous? In this section, we address this issue by investigating the subtle evolution of the geometry of local packing at the neighbor scale, $k \sim 2\pi/\overline{\sigma}$. In particular, we confirm a mild increase of bond orientational order upon increasing density (Sec. 5.1) and reveal the emergence of distorted icosahedral local structures (Sec. 5.2). Above the onset density, these structural features are present in both fluid and jammed states.

## 5.1  Bond-orientational order

Bond-orientational order (BOO) parameters [74] have been used extensively to characterize the local structure of jammed packings [15,21,36]. The idea is to compute rotational invariants from a multiple expansion of the nearest neighbors distance distribution, and compare the results to the values observed for known crystal structures [74]. Inspection of the combined distribution of BOO parameters of different orders allows one to disentangle structures characterized by different local symmetries [75]. For a given particle $i$, the local BOO parameter is defined as

$$Q_{l,i} = \sqrt{\frac{4\pi}{2l+1} \sum_{m=-l}^{l} \left| \frac{1}{n_b(i)} \sum_{j=1}^{n_b(i)} Y_{l,m}(\mathbf{r}_{ij}) \right|^2 }, \tag{2}$$

where $n_b(i)$ is the number of nearest neighbors of the $i$-th particle, $Y_{l,m}(\mathbf{r}_{ij})$ is the spherical harmonic of degree $l$ and order $m$, and $\mathbf{r}_{ij}$ is the vector distance between particle $i$ and $j$. The average BOO parameters, $Q_l = (1/N)\sum_i Q_{l,i}$, can then be used to characterize the local structure of a packing.

It is well known that the values of local BOO parameters in dense, disordered packings depend sensitively on the definition of the neighbors network surrounding each particle [19]. In disordered packings, neighbors are often defined as particles whose distance contributes to the first peak of the radial distribution function; alternatively, the nearest neighbors network is obtained from a Voronoi tessellation of the particles' coordinates [76]. In order to cure the sensitivity of BOO parameters to the details of neighbors network, Mickel *et al.* [19] have proposed to weight each bond entering the calculation of $Q_{l,i}$ by the area of the corresponding face of the surrounding Voronoi cell. The resulting weighted BOO parameters are then defined

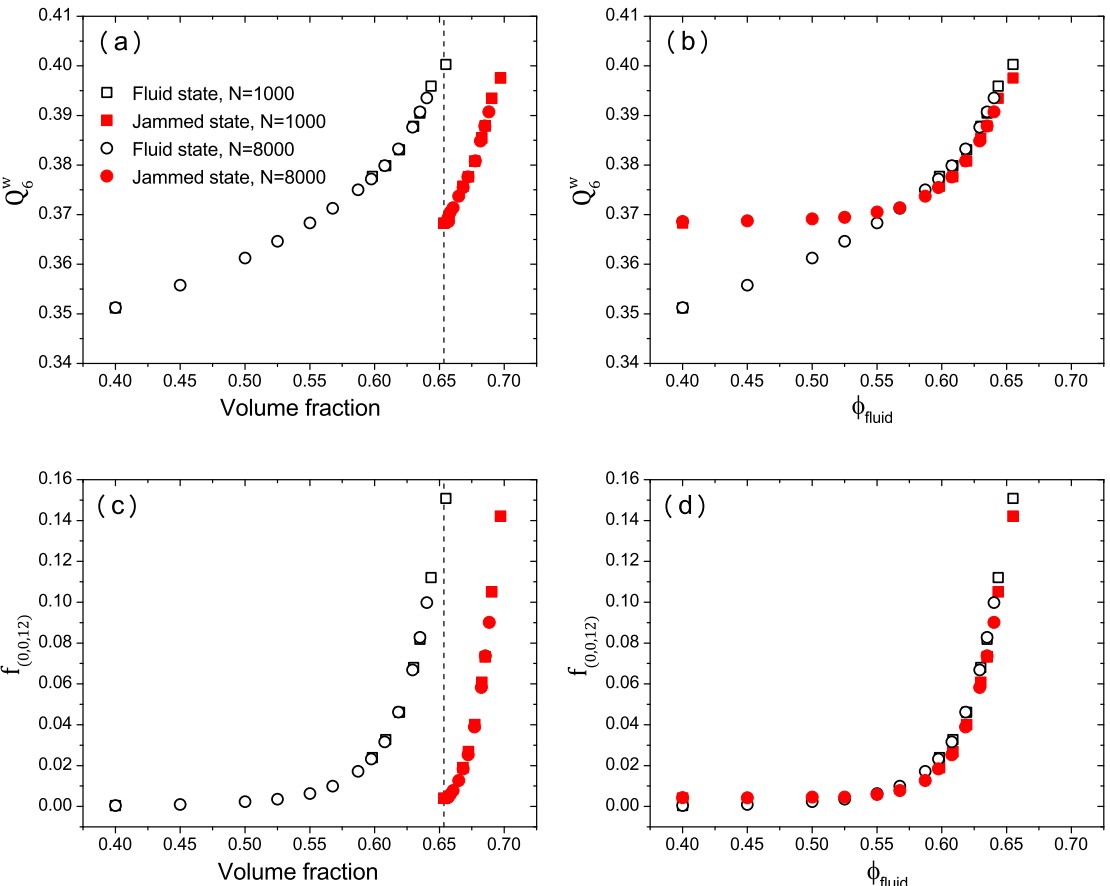

Figure 4: (a, b): Bond orientational order parameter $Q_6^w$ for the fluid and jammed states as a function of their volume fraction (a), or the volume fraction of the parent fluid, $\phi_{\text{fluid}}$, (b). (c, d): Similar plots for the fraction of icosahedral structures characterized by the $(0, 0, 12)$ Voronoi signature. The vertical lines are a guide for the eye.

as

$$Q_{l,i}^w = \sqrt{\frac{4\pi}{2l+1} \sum_{m=-l}^{l} \left| \sum_{j=1}^{n_b(i)} \frac{A_{ij}}{A_i} Y_{l,m}(\mathbf{r}_{ij}) \right|^2}, \tag{3}$$

where $A_{ij}$ denotes the area of the face of Voronoi cell connecting particle $i$ and $j$ and $A_i = \sum_j A_{ij}$. A similar idea had been suggested long ago in Ref. [74], to cure the artifacts mentioned above. In the following, we compute the weighted BOO employing the neighbors network determined via a radical Voronoi tessellation [76] obtained using Voro++ [77].

A standard indicator of structural ordering is the average $Q_6$, which takes large values for ordered, close-packed structures ($Q_6 = Q_6^w = 0.575$ for the fcc cell and $Q_6 = Q_6^w = 0.663$ for the icosahedron). In Figs. 4(a, b), we show the weighted BOO parameter $Q_6^w$ for both equilibrium fluid (open symbols) and jammed (closed symbols) states. The data are shown as a function of the corresponding volume fraction of jammed states, $\phi_J$, and as a function of the volume fraction of the parent equilibrium fluid, ($\phi_{\text{fluid}}$), in panels (a) and (b), respectively. In both fluid and jammed states, $Q_6^w$ increases monotonically with increasing packing fraction, indicating a systematic, progressive local ordering along the J-line. In agreement with the results of Ref. [19], we found that the values of the standard $Q_6$, obtained using the Voronoi neighbors network are smaller than for its weighted counterpart, $Q_6^w$ (by 10-15% in our case,

result not shown).

The different representations of the data in Fig. 4 convey two distinct messages: Panel (a) shows that fluid and jammed states can have a similar volume fraction and yet clearly different local structure, as illustrated by the vertical dashed line around $\phi_J = 0.655$ [44, 78]. Panel (b) demonstrates instead that above the onset volume fraction, $\phi_{\mathrm{onset}} \simeq 0.56$, around which glassy dynamics starts to manifest in the equilibrium fluid [79], the equilibrium fluid and the corresponding jammed states are characterized by very similar bond orientational order; at the local scale, jammed configurations retain essentially the local structure of the parent fluid. This result is confirmed by analysis of the other local structure metrics detailed below.

The evolution of $Q_6^w$ is generally not enough to detect the presence of local crystal structures in dense packings. In particular, it is difficult to disentangle local fcc structures from distorted icosahedra [19], which both give similar $Q_6$ values. To resolve fine details of the local structure, one has to resort to scatter plots of rotational invariants of different orders [74], and possibly introduce an additional averaging procedure over the neighbors [80]. Mickel *et al.* [19] have suggested to measure $Q_2^w$ invariants, which are zero for most crystal structures as a simpler approach to quantify the degree of local crystalline order. We found that $Q_2^w$ decreases mononotically by increasing volume fraction for both fluid and jammed configurations, and that its variation is smooth and continuous (not shown). Typical values of the average $Q_2^w$ range from about 0.1 at $\phi = 0.4$ to 0.05 at the largest packing fractions. These values are consistent with those observed in non-crystalline packings of monodisperse hard spheres [19]. This, together with the smooth evolution of the fluid equation of state and the regularity of the partial structure factors at small $k$ [79], confirms that our packings are fully amorphous along the whole J-line.

## 5.2 Locally favored structure

A slight but systematic change of bond orientational order along the J-line has been noted before [34, 36], albeit over a smaller range of volume fractions. While this suggests some sort of ordering of the packings, uncovering the precise nature of local order remains difficult and requires knowledge of a dictionary of "known" reference structures. In this section, we employ a strategy based on the statistics of Voronoi cell shapes [81, 82] which does not rely on any a priori reference to specific motifs. This approach has been employed successfully in the context of glasses, see e.g. Refs. [46, 83] for recent reviews, but its use in the context of jammed packings remains very limited [20].

As in Sec. 5.1, we perform a radical Voronoi tessellation [76] of each configuration. In this construction, which duly accounts for the size polydispersity, each particle $i$ in the system is enclosed in a polyhedral cell such that all points whose tangent distance from the surface of particle $i$ is smaller than the tangent distance from the surface of particle $j$ with $j \neq i$ [76]. The shape of a Voronoi cell encodes detailed information about the local arrangements of the neighbors around the central particle, hence the local structure. We characterize the shape of a cell through its signature $(n_3, n_4, n_5, ...)$, where $n_q$ is the number of faces of the cell with a given number $q$ of vertices.[1] By averaging over an ensemble of configurations, we detect the most frequent signatures and monitor their concentration as a function of the relevant control parameter (here the volume fraction). This approach provides a fairly robust assessment of the preferred local order, often termed "locally favored structure" in simple models of glasses [46].

We find that at low volume fractions the fluid exhibits a variety of different Voronoi signatures, typical of highly disordered fluids. Upon increasing $\phi_{\mathrm{fluid}}$, however, fluid samples become increasingly rich in $(0, 0, 12)$ signatures, which are associated to icosahedral local order. This kind of signature is by far the most frequent one at largest volume fractions. In

---

[1]We disregard null values of $n_q$ for $q > q'$, where $q'$ is the largest number of vertices such that $n_{q'} > 0$.

| $\phi_J = 0.656$ | | $\phi_J = 0.678$ | | $\phi_J = 0.697$ | |
|---|---|---|---|---|---|
| (0,2,8,1) | 2.8% | (0,2,8,1) | 5.1% | (0,0,12) | 14.2% |
| (0,3,6,4) | 2.5 | (0,2,8,2) | 4.4 | (0,1,10,2) | 7.0 |
| (0,3,6,3) | 2.4 | (0,2,8) | 4.1 | (0,2,8) | 6.3 |
| (0,2,8,2) | 2.3 | (0,0,12) | 3.9 | (0,2,8,1) | 5.7 |
| (0,3,6,1) | 2.0 | (0,1,10,2) | 3.7 | (0,2,8,2) | 4.6 |
| (0,4,4,3) | 1.9 | (0,3,6,4) | 3.2 | (0,3,6) | 3.7 |
| (0,2,8) | 1.8 | (0,3,6,3) | 2.8 | (0,3,6,1) | 3.0 |
| (0,3,6,2) | 1.7 | (0,3,6,1) | 2.8 | (0,3,6,4) | 2.4 |
| (0,3,6) | 1.6 | (0,3,6) | 2.3 | (0,3,6,3) | 2.1 |
| (1,2,5,3) | 1.4 | (0,2,8,4) | 2.1 | (0,2,8,4) | 2.0 |

Table 1: Most frequent Voronoi signatures and the corresponding average percentages in jammed configurations. The corresponding parent fluid packing fractions are $\phi_{\text{fluid}} = 0.500$ ($N = 8000$), 0.618 ($N = 8000$), and 0.655 ($N = 1000$), respectively.

Figs. 4(c, d), we show the fraction $f_{(0,0,12)}$ of icosahedral structures for both fluid and jammed states. As for panels (a) and (b), we plot the data as a function of $\phi_J$ and of $\phi_{\text{fluid}}$. We find that $f_{(0,0,12)}$ for both fluid and jammed states is essentially negligible below $\phi_{\text{fluid}} \simeq 0.56$, but then it increases rapidly with increasing volume fraction to reach a value of about 14% for our densest packings. We note that if we also take into account all particles connected to centers of icosahedral structures, as done for instance in Ref. [84], this fraction reaches about 80 % at the largest density. Superficially, this behavior resembles the one found in the binary Lennard-Jones mixture introduced in Ref. [85], for which $f_{(0,0,12)}$ shows a marked increase around the onset of slow dynamics [86].

Turning our attention to jammed packings, no distinguishable local order is detectable below $\phi_{\text{onset}}$. Figure 4(d) demonstrates that the sudden emergence of the icosahedral local order around $\phi_{\text{onset}}$ in the equilibrium fluid is, again, inherited by the jammed packings. It is thus tempting to attribute the increase of $\phi_J$ along the J-line to the appearance of distorted icosahedral arrangements, which help maintaining an overall amorphous organization of the packings [20]. In Table 1, we report the occurrences of the most frequent Voronoi signatures in jammed packings at selected volume fractions. We note that, on average, (0,0,12) cells have contact numbers $Z$ slightly smaller than 6 ($Z = 5.4$). We found that some less abundant signatures, such as (0,3,6) and (0,2,8), are associated to even smaller values of $Z$. These trends suggest a correlation between $Z$ and the number of nearest neighbors, which we tentatively attribute to the size dispersity of the packings.

The presence of icosahedral order and, more generally, polytetrahedral order in fluid and jammed hard sphere packings has been discussed before, see *e.g.* Refs [17, 20, 87, 88]. The results presented above differ from earlier findings on two important aspects. First, the amount of $(0, 0, 12)$ signatures found in our polydisperse packings substantially exceeds the one reported in monodisperse jammed packings [20] and is also appreciably larger than the one reported in less polydisperse hard sphere at equilibrium [88]. However, because of the large size polydispersity, the local arrangements associated to $(0, 0, 12)$ signatures are often distorted and irregular. Such issues are usually neglected in the analysis of icosahedral order in simple binary mixtures [86].

To quantify the degree of asphericity of Voronoi cells, we analyzed the distribution of distances $r_{ij} = |\mathbf{r}_i - \mathbf{r}_j|$ separating a central particle $i$ to its neighbors $j$. As a simple measure of cell

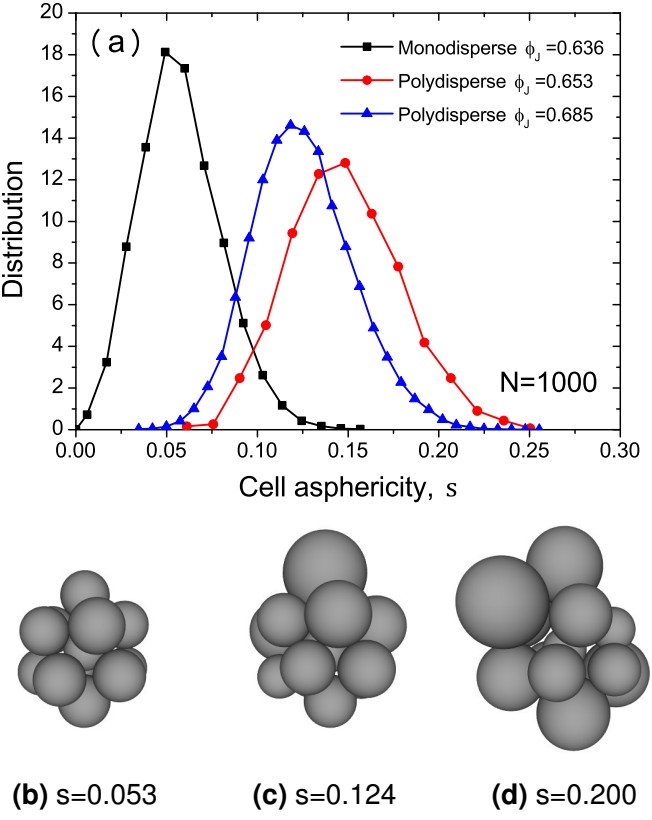

**(b)** s=0.053    **(c)** s=0.124    **(d)** s=0.200

Figure 5: (a): Distribution of asphericity parameter $s$ for polydisperse hard spheres for two different volume fractions, and for a monodisperse hard sphere system. (b, c, d): Typical icosahedral structures with varying degree of asphericity $s$.

aspherity, we compute the normalized standard deviation of the distances from particle $i$,

$$s_i = \frac{1}{\tilde{r}_i} \sqrt{\frac{1}{n_b(i)} \sum_{j=1}^{n_b(i)} (r_{ij} - \tilde{r}_i)^2},$$

where $\tilde{r}_i = (\sum_{j=1}^{n_b(i)} r_{ij})/n_b(i)$ is the average nearest neighbors distance of particle $i$. See Ref. [89] for a more systematic approach based on Minkowski tensors. Of course $s \geq 0$ and the equality holds for a perfectly regular structure. In Fig. 5(a) we show the probability distribution of the asphericity parameter, $P(s)$, of icosahedral structures for jammed packings for a low ($\phi_J = 0.653$) and a high ($\phi_J = 0.685$) packing fraction. We find that the distribution is not very sensitive to the value of $\phi_J$. For comparison, we also include results for $P(s)$ measured in monodisperse hard sphere jammed packings. We obtained the latter by compressing Poisson distributed configurations at $\phi_{fluid} = 0.3$ using the algorithm described in Sec. 2.3. On average, icosahedral structures detected in polydisperse packings are more aspherical by about a factor two than those found in the monodisperse hard spheres. We found that the asphericity of icosahedra in a glassy binary mixture with modest size ratio [85], for which icosahedral order is most pronounced [45], is intermediate between those of the two systems in Fig. 5(a). Figures 5(b-d) show representative icosahedral structures detected in our densest packings. They are representative of different degrees of asphericity, ranging from fairly regular (b) to intermediate (c) and highly irregular (d) structures.

These results indicate that, despite the abundance of $(0, 0, 12)$ Voronoi cells, the nature of icosahedral order in polydisperse packings might be different from the one observed in

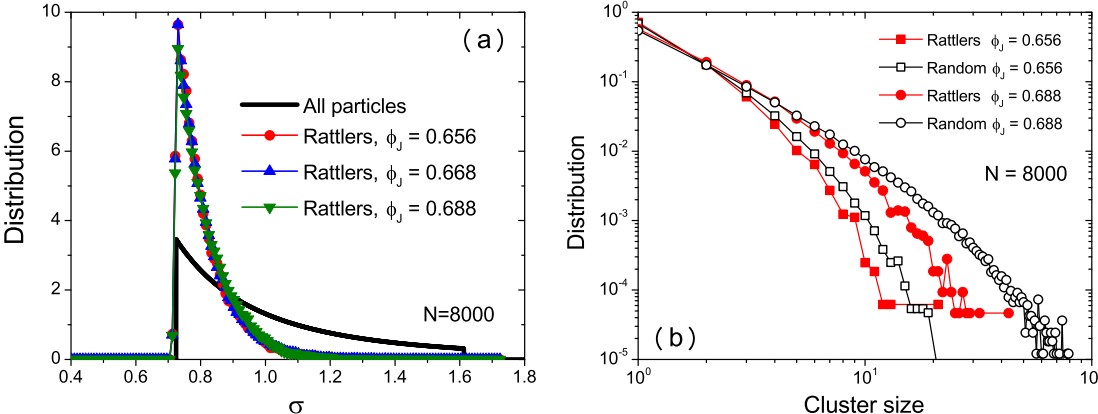

Figure 6: (a) Diameter distribution for all particles, $f(\sigma)$, and diameter distribution for rattlers only, $f_R(\sigma)$. (b) Cluster size distribution for the rattlers (closed symbols), compared to the same fraction of randomly selected particles in the same packings (open symbols).

simple binary mixtures. In fact, icosahedral structures in highly polydisperse spheres are more distorted and inherently more irregular due to the increased compositional freedom. In binary mixtures with sufficiently small size ratio icosahedral structures can strongly influence the local mobility of the particles in the supercooled regime [45, 84, 90]. Further work is needed to assess the relevance of icosahedral local order in dense, highly polydisperse packings, in particular in the context of glass transition studies.

## 5.3 Spatial organization of rattlers

In Fig. 2(b) we have shown that the fraction of rattlers grows steadily with increasing $\phi_J$. Naively, one might expect rattlers to occupy a larger free space, thus a large number of rattlers seems at odds with the increased efficiency of the packings.

First, we assess the size distribution of rattlers, $f_R(\sigma)$, along the J-line, see Fig. 6(a). By comparing $f_R(\sigma)$ to the distribution of all particles, $f(\sigma)$, we find rattlers are mostly small particles, which is an intuitive result. Interestingly, the shape of $f_R(\sigma)$ hardly changes along the J-line, whereas the fraction of rattlers increases substantially.

Next, we inspect the spatial organization of rattlers. To get a qualitative idea of their real space structure, we show in Figs. 7 some typical snapshots of rattlers found in jammed packings at $\phi_J = 0.656$ and $\phi_J = 0.688$. Rattlers appear to be distributed in rather homogeneous way in this representation. To get a more quantitative picture, we follow previous work [91] and study the spatial organization of rattlers into clusters. We define a rattler cluster as a group of rattlers where each rattler is neighbor to at least another member of the cluster. As in the previous sections, neighbors are identified through a radical Voronoi tessellation. Note that our analysis differs from the one of Ref. [91], in which rattler clusters were classified according to the connectivity of their corresponding cages.

It is sometimes assumed that rattlers are randomly distributed in the system [92]. To test this idea, we randomly pick a fraction of particles with the same concentration as the rattlers and compute the corresponding cluster size using the same definition given above. We show the distribution of rattler cluster sizes for jammed states with a low ($\phi_J = 0.656$) and a high ($\phi_J = 0.688$) volume fraction in Fig. 6(b). We find that the rattler clusters are smaller than the one formed by the randomly chosen particles, for both low and high $\phi_J$ values. This implies that overall the positions of the rattlers are only weakly correlated, displaying a slight tendency to form small clusters [91]. This is confirmed by visual inspection of snapshots of jammed

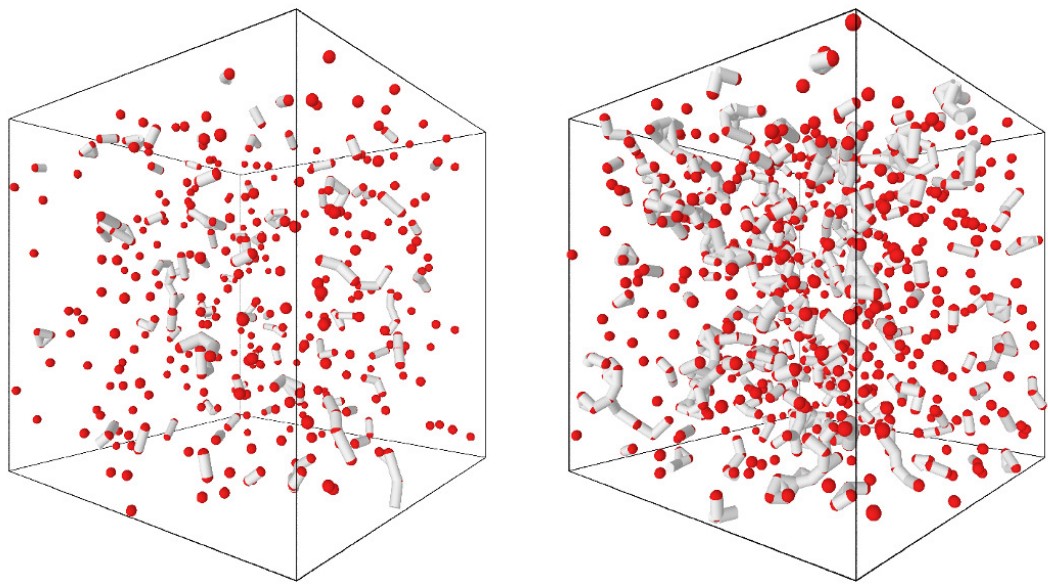

Figure 7: Typical snapshots of rattlers (red spheres) in jammed packings at $\phi_J = 0.656$ (left) and $\phi_J = 0.688$ (right). The white bonds connect neighboring rattlers that form a cluster.

packings along the J-line, see Fig. 7 for representative examples. These relatively compact clusters might lead to small volume fraction fluctuations and may also affect hyperuniform behavior, as discussed in Sec. 7.

## 6 Structure at the large scale

Here we consider the structure of the packings at the large scale. The corresponding wave number regime is $k \to 0$, which means that we quantify fluctuations at very large length scales.

### 6.1 Hyperuniformity

Equilibrium fluids are characterized by a finite value of the static structure factor $S(k)$ at $k \to 0$, which is associated to a finite isothermal compressibility. However, it is known that hard sphere packings at the jamming transition show "unexpected" (that is, more complex) density fluctuations [22, 93]. It was reported that $S(k)$ in monodisperse jammed packings obeys a surprising linear behavior at low $k$, $S(k) \propto k$, which characterizes hyperuniform materials [22, 93]. It is sometimes assumed that hyperuniformity is related to the criticality of the jammed states [47], but the nature of this criticality and the connection to other signatures of jamming remain unclear [25, 26].

When the system is a mixture of several components or is continuously polydisperse, the structure factor $S(k)$ of jammed states does not show a linearly vanishing behavior. Instead, $S(k)$ takes a finite value at $k \to 0$ just as in fluid states [94, 95]. We show $S(k)$ of our polydisperse system for both fluid and jammed states in Figs. 8(a, b). These results confirm that the shape of $S(k)$ of jammed states is similar to the fluid ones, taking in particular a finite value, $S(k \to 0) \simeq 0.5$, for all jammed states along the J-line, with no sign of a vanishing signal at low $k$.

In Refs. [23, 92], the concept of hyperuniformity was generalized from density to volume fraction fluctuations to discuss the hyperuniformity of multi-components system and of

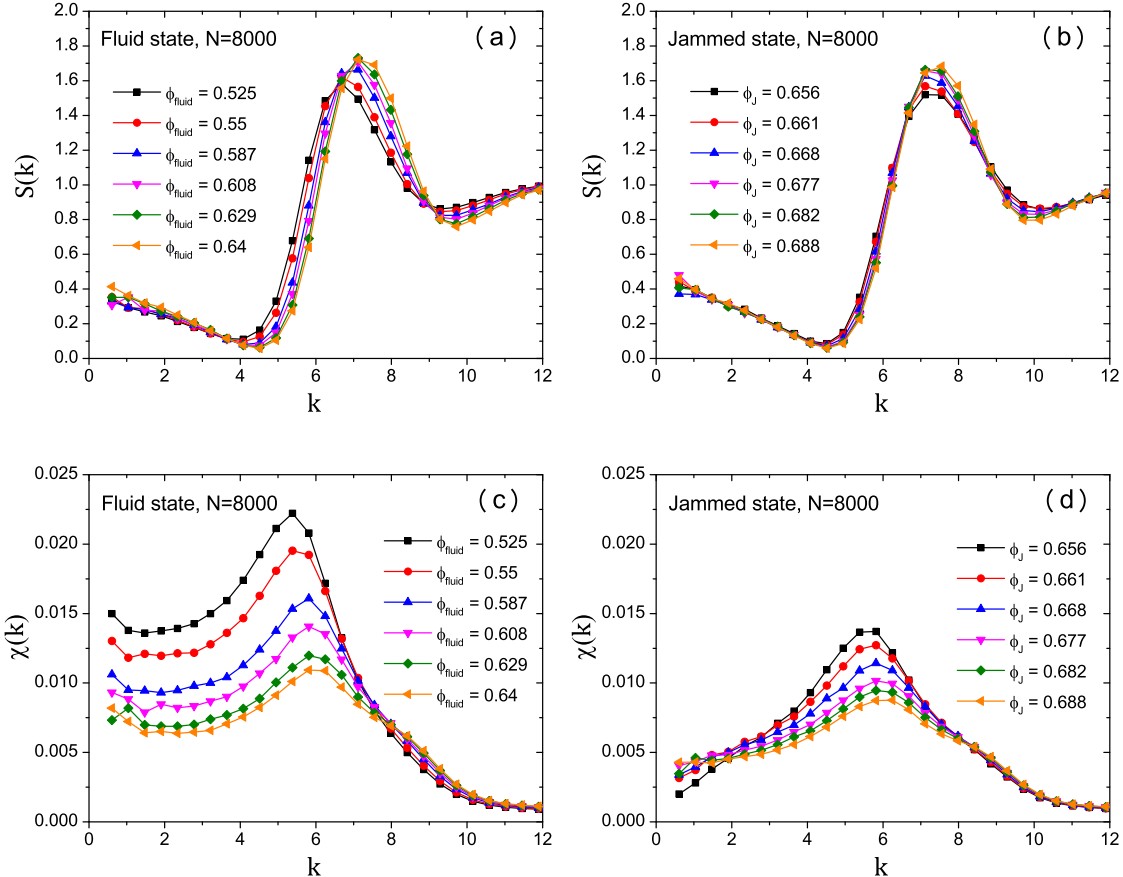

Figure 8: (a, b): Static structure factor $S(k)$ reflecting number density fluctuations for fluid (a) and jammed (b) states. (c, d): Spectral density $\chi(k)$ reflecting volume fraction fluctuations for fluid (c) and jammed (d) states. Nearly hyperuniform behavior expected for $\chi(k)$ is only observed for the lowest $\phi_J$ value in (d).

particles with non-spherical shapes. Thus, an appropriate observable to characterize the hyperuniformity of polydisperse systems is the spectral density $\chi(k)$, which quantifies the volume fraction fluctuations [23, 92]. It is defined as

$$\chi(\mathbf{k}) = \frac{1}{V} \langle I_{\mathbf{k}} I_{-\mathbf{k}} \rangle, \tag{4}$$

where $I_{\mathbf{k}}$ is the Fourier transform of the indicator function $I(\mathbf{r})$. For spherical particle systems, $I(\mathbf{r})$ is given by

$$I(\mathbf{r}) = \sum_{i=1}^{N} \theta(R_i - |\mathbf{r} - \mathbf{r}_i|), \tag{5}$$

where $R_i$ is the radius of the $i$-th particle, $R_i = \sigma_i/2$. For homogeneous isotropic systems in $d = 3$, $I_{\mathbf{k}}$ becomes

$$I_{\mathbf{k}} = \sum_{i=1}^{N} \frac{4\pi}{k^3} \left[ \sin(kR_i) - (kR_i)\cos(kR_i) \right] e^{-i\mathbf{k}\cdot\mathbf{r}_i}. \tag{6}$$

Note that alternative definitions of the volume fraction fluctuations in the reciprocal space can be applied to study hyperuniformity in jammed packings [24, 96].

In Figs. 8(c, d), we show our numerical results for $\chi(k)$ for both fluid and jammed states. Before discussing $\chi(k)$ of jammed states at small $k$, where signs of hyperuniformity may be

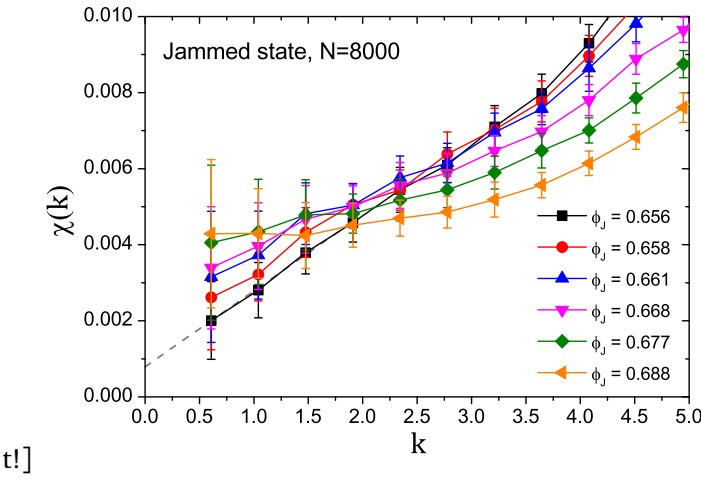

t!]

Figure 9: Zoom in the small $k$ region of the spectral density $\chi(k)$ for jammed states shown in Fig. 8(d). The error bars represent the standard deviation obtained from sample to sample fluctuations. The gray dashed line is a linear fit for $\phi_J = 0.656$, which extrapolates to $\chi(k \to 0) = 7.9 \times 10^{-4}$ for the lowest $\phi_J$. Instead $\chi(k \to 0) = 4.2 \times 10^{-3}$ for the largest $\phi_J$.

found, it is instructive to study $\chi(k)$ for the fluid states and its volume fraction dependence. We find that $\chi(k)$ for fluid states has a broad peak near $k \sim 2\pi/\overline{\sigma}$ and a flat plateau at smaller $k$ [97]. In addition, the overall amplitude of $\chi(k)$ decreases as the volume fraction $\phi_{\text{fluid}}$ increases at $k \lesssim 2\pi/\overline{\sigma}$, reflecting that volume fraction fluctuation are suppressed in order to achieve denser particle packings in equilibrium. This observation contrasts with the well-known increase of the first peak of $S(k)$ with increasing the density in dense fluids [98] which instead reflects the slight increase in local scale structural correlations as the density is increased.

In jammed states, the peak height of $\chi(k)$ is much smaller than in corresponding fluid states, and the peak amplitude decreases as $\phi_J$ increases, reflecting again that further optimization of local configurations has been realized. Note that $\chi(k)$ displays a rather strong volume fraction dependence in both fluid and jammed states, which contrasts with the relatively weak density dependence of $S(k)$ over the same range. This insensitivity of $S(k)$ in supercooled liquids is traditionally taken as a hallmark of glass physics [98]. However, our results demonstrate that $\chi(k)$ is a more sensitive probe of structural changes at the level of two-point correlations. This suggests that $\chi(k)$ might be a good observable to characterize the equilibrium structure of dense hard sphere systems.

Having described the main features of $\chi(k)$, we now concentrate on the low-$k$ behavior in jammed states. A zoom in this region is shown in Fig. 9, to investigate more carefully the possible existence of a hyperuniform behavior in this regime. At the lowest $\phi_J$ along the J-line for $N = 8000$, $\phi_J = 0.656$, $\chi(k)$ linearly decreases with decreasing $k$, as observed in previous numerical works [23, 24, 47, 92]. By fitting this linear behavior (shown with the dashed line), we can extrapolate the limit $k \to 0$ and we obtain a small finite value, $\chi(k \to 0) = 7.9 \times 10^{-4}$ for this volume fraction. Therefore, one might conclude that the system at $\phi_J = 0.656$ is very close to being hyperuniform [47]. However, as $\phi_J$ is increased from its lowest value, $\chi(k)$ quickly deviates from this linear behavior, and it even becomes flat, within our error bars, at low $k$. Simultaneously, $\chi(k \to 0)$ obtained by linear extrapolation increases systematically from $\chi(k \to 0) \approx 7.9 \times 10^{-4}$ to $\chi(k \to 0) \approx 4.2 \times 10^{-3}$ with increasing $\phi_J$, suggesting that deviations from hyperuniformity become stronger when $\phi_J$ increases.

Recently, deviations from hyperuniformity were reported in both two [24] and three [25, 26] spatial dimensions. Both works employed rapid quenches from fully random configura-

tions using energy minimization protocols. These studies showed that $\chi(k)$ (and thus $S(k)$ in monodisperse systems) linearly decreases with decreasing $k$, but it always saturates at a certain $k^*$ ($k^* \sim 0.2$ in Ref. [24] and $k^* \sim 0.4$ in Refs. [25, 26]), indicating that hyperuniformity is eventually avoided at sufficiently small $k$. Ref. [47] attributes the breakdown of hyperuniformity at very small $k$ to a lack of numerical accuracy in conventional jamming algorithms. It is argued that producing truly jammed, hyperuniform states is a highly difficult numerical task and thus the observed saturation at $k^*$ using energy minimization could stem from numerical inaccuracy of a given algorithm.

However, the systematic deviation of $\chi(k)$ at denser $\phi_J$ of our system shown in Fig. 9 presumably has a different origin, because the system size we use, $N = 8000$, is not large enough to detect the saturation at $k^*$ reported in previous work. Also, the systematic increase of $\chi(k \to 0)$ is observed for a given system size and within a given numerical algorithm. Therefore, the strong deviations from hyperuniformity reported here must have a physical, rather than a computational, origin.

Our jammed packings at various $\phi_J$ are obtained with a similar degree of precision, in particular isostaticity is equally well satisfied for all configurations along the J-line. However, they display an increasing tendency to depart from hyperuniformity as $\phi_J$ is increased. Therefore, we conclude that the jamming criticality, which is well-obeyed all along the J-line, and hyperuniformity, which is increasingly violated, are distinct concepts. This conclusion is consistent with the fact that the linear dependence of $\chi(k)$ (and $S(k)$), which might be interpreted as a "precursor" of hyperuniformity [99], is robustly observed at densities well below [99] and above [24, 25] $\phi_J$ and even in the presence of thermal fluctuations [25], whereas the jamming criticality originating from isostaticity is very quickly erased in similar conditions [100].

Furthermore, the nearly hyperuniform behavior observed at the lowest $\phi_J$ in this work implies that hyperuniformity, if it were to be observed, should characterize the lowest end of the J-line [32, 40]. Although the "maximally random jammed state" [2] and the lowest end point of the J-line are conceptually distinct, our observations lead us to speculate that these two concepts may be identical [101], and may both display the strongest (although presumably still imperfect) signature of hyperuniformity. This conclusion, based on our numerical findings, also directly contradicts an opposite theoretical prediction recently made in Ref. [102].

A hypothesis that could explain the present observations is that the increasing deviation from hyperuniformity observed on increasing $\phi_J$ correlates with the increasing number of rattlers in the corresponding packings. Rattlers can be seen as "defects" in the volume fraction field, and a finite fraction of rattlers could induce a finite amount of volume fraction fluctuations at large scale. This hypothesis is hard to test directly, as we cannot independently vary the fraction of rattlers and the degree of hyperuniformity in hard sphere packings. We notice that the fraction of rattlers increases by only a factor of 2 while the limit $\chi(k \to 0)$ increases by about an order of magnitude, which suggests that rattlers may not be the central explanation for this observation. This conclusion was indirectly tested in Ref. [24], which showed that upon compression of jammed packings the number of rattlers decreased but the behavior of $\chi(k)$ was essentially unchanged. We also confirmed this behavior in our simulations (not shown). We will discuss this issue further in Section 7.

## 6.2 Finite-size fluctuations of the critical density of jamming

Second order phase transitions are usually associated with diverging length scales [103]. Finite-size scaling represents a powerful tool to understand the nature of such transitions and of the corresponding diverging length scales. Although several distinct important length scales have been identified for the jamming transition [7, 25, 99, 100, 104–110], the particular length scale responsible for the finite-size effects of $\phi_J$ is not well understood [5, 105, 111].

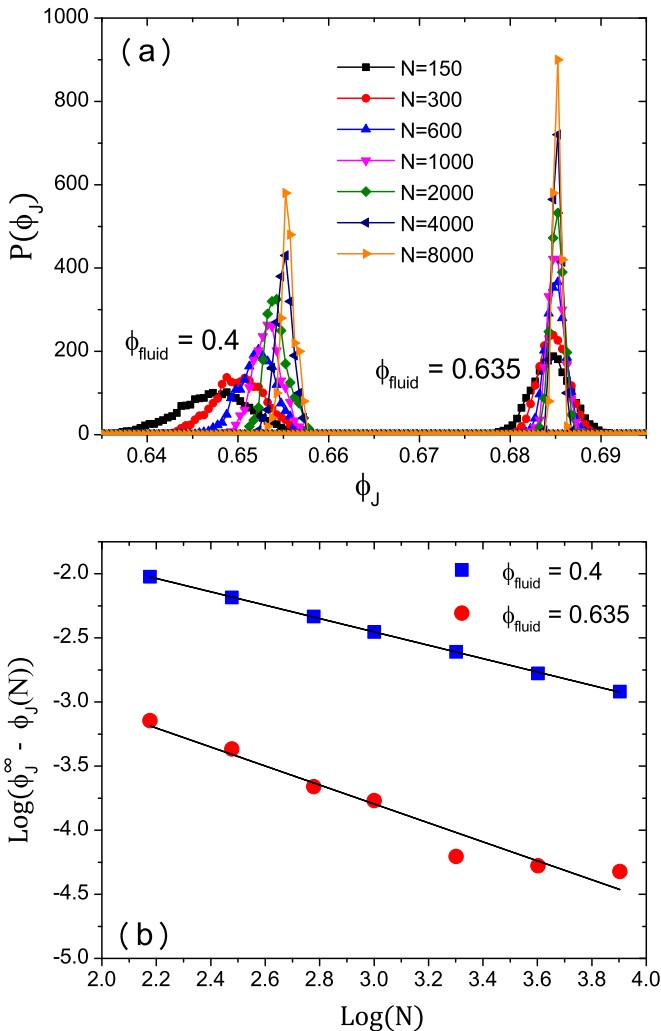

Figure 10: (a): Probability distribution function of $\phi_J$ for several $N$ obtained by compressing dilute ($\phi_{\text{fluid}} = 0.4$) and dense ($\phi_{\text{fluid}} = 0.635$) equilibrium fluid configurations. (b): Finite-size scaling plot according to Eq. (7). Finite-size effects on the quantity $\phi_J(N)$ are considerably suppressed for $\phi_{\text{fluid}} = 0.635$.

A qualitatively different aspect of the jamming transition compared to usual critical phenomena is the fact that the transition point is not unique and is protocol-dependent [11]. Thus, we examine the finite-size effect of $\phi_J$ for two different protocols which produce different averaged $\phi_J$ values. In practice, we compare results of two protocols, the non-equilibrium compression from dilute ($\phi_{\text{fluid}} = 0.4$) and dense ($\phi_{\text{fluid}} = 0.635$) fluids, which produce low ($\phi_J \simeq 0.657$) and high ($\phi_J \simeq 0.685$) jamming volume fractions. We expect a smooth evolution of the behavior with $\phi_J$, but these measurements are numerically demanding, so we limit ourselves to only two different critical points on the extremes of the J-line.

We show the probability distribution function $P(\phi_J)$ for several system sizes $N$ in Fig. 10(a). For the compression of the dilute fluid, $\phi_{\text{fluid}} = 0.4$, a broad Gaussian shape of $P(\phi_J)$ is obtained for the smallest system size. As $N$ increases, the distribution becomes narrower and the position of the peak shifts to higher volume fractions, as reported previously [5, 60, 112].

For the compression of the dense fluid, $\phi_{\text{fluid}} = 0.635$, the width of the distribution for a given $N$ is slightly narrower than the one for $\phi_{\text{fluid}} = 0.4$ [14, 34], but the distribution again becomes more sharply peaked as $N$ increases. Remarkably, the peak position hardly shifts with

$N$, in strong contrast with $\phi_{\mathrm{fluid}} = 0.4$. Therefore, for $\phi_{\mathrm{fluid}} = 0.635$, the finite size effect in terms of the mean value of $\phi_{\mathrm{J}}$ is significantly suppressed and $\phi_{\mathrm{J}}(N)$ quickly approaches the thermodynamic value $\phi_{\mathrm{J}}^{\infty}$. For both protocols, the latter value can be extracted using standard finite-size scaling,

$$\phi_{\mathrm{J}}(N) = \phi_{\mathrm{J}}^{\infty} - \delta N^{-1/\nu d}, \tag{7}$$

where $\delta$ and $\nu$ are fitting parameters, and $d$ is the number of spatial dimensions. Note that we use the mean value for $\phi_{\mathrm{J}}(N)$ instead of the peak value [5], since we do not have enough statistics to determine the peak position with high accuracy. The results of this fit are shown in Fig. 10(b). We obtain $\phi_{\mathrm{J}}^{\infty} = 0.65698 \pm 0.00009$ and $\nu = 0.64 \pm 0.01$ for $\phi_{\mathrm{fluid}} = 0.4$, and $\phi_{\mathrm{J}}^{\infty} = 0.68548 \pm 0.00002$ and $\nu = 0.41 \pm 0.04$ for $\phi_{\mathrm{fluid}} = 0.635$, respectively. The obtained $\nu$'s are slightly different from previous reports [5, 113]. Note that $\nu$ for $\phi_{\mathrm{fluid}} = 0.4$ is compatible with very recent numerical work [102]. However, we do not wish to discuss these values quantitatively because $\nu$ might be protocol or algorithm dependent, and corrections to scaling should be included before drawing any strong conclusions [111]. We do not treat these corrections to scaling because they require huge statistics. Our main result here is more qualitative; finite size effects on $\phi_{\mathrm{J}}$ are significantly suppressed for the compression from the dense fluid. The prefactor $\delta$ in Eq. (7) is more than 10 times smaller for the compression of the dense fluid, as can be directly seen in the data shown in Fig. 10(b). The smallness of the finite-size effect in fact makes a quantitative determination of the critical exponent very difficult for the largest $\phi_{\mathrm{J}}$ values [36]. Our data conclusively demonstrate that the J-line is not due to a finite size effect and that it remains instead well-defined in the thermodynamic limit.

To get more physical insights into the finite-size effects for $\phi_{\mathrm{J}}$, we monitor the probability distribution function $P(\Delta r)$ of single particle displacements during the compression from the fluid to the jammed states, $\Delta r = |\mathbf{r}_i^{\mathrm{jam}} - \mathbf{r}_i^{\mathrm{fluid}}|$, where $\mathbf{r}_i^{\mathrm{fluid}}$ and $\mathbf{r}_i^{\mathrm{jam}}$ are the positions of particle $i$ in the equilibrium parent fluid configuration and in the corresponding jammed configuration, respectively. The distribution $P(\Delta r)$ is computed for all the particles (including rattlers). $P(\Delta r)$ quantifies how much the particles need to move or rearrange during the compression until the system is jammed. In Figs. 11(a, b), we show $P(\Delta r)$ as a function of either $\Delta r$ or $(\Delta r)^2$, for three different system sizes, $N = 150$, 1000, and 8000. For $\phi_{\mathrm{fluid}} = 0.4$, $P(\Delta r)$ has a rather long tail, indicating that the particles may perform large displacements before finding the final jammed configuration. This tail can be fitted by a Gaussian form, $P(\Delta r) \propto \exp[-b(\Delta r)^2]$, where $b$ is a constant [see Fig 11(b)], which suggests that there is no strong positional correlation between before and after compression. Importantly, a noticeable finite-size effect is found in these tails, in the sense that smaller systems are characterized by smaller particle displacements. This observation provides a possible explanation for the observed finite-size effect in $\phi_{\mathrm{J}}$: particles in smaller systems do not explore space as they do in large systems and thus get jammed in less optimized (*i.e.* less dense) configurations, leading to smaller $\phi_{\mathrm{J}}$, as observed in Fig. 10.

Interestingly, for $\phi_{\mathrm{fluid}} = 0.635$, the tail in $P(\Delta r)$ is significantly suppressed compared to the one for $\phi_{\mathrm{fluid}} = 0.4$, indicating that particles get jammed in a position that is actually very close to their original position in the parent fluid. This observation is of course consistent with the fact that the parent fluid and its corresponding jammed state display very similar geometric structure when $\phi_{\mathrm{J}}$ is large, as shown above in Figs. 4(b, d). Simultaneously, the finite-size effect on the tail in $P(\Delta r)$ almost disappears, which parallels the suppression of the finite-size effect in $\phi_{\mathrm{J}}$ values. Furthermore, $P(\Delta r)$ is no longer Gaussian, but exhibits an exponential tail, $P(\Delta r) \propto \exp[-a\Delta r]$, where $a$ is a constant. This functional form has been reported in supercooled liquids undergoing a transition between nearby inherent structures [114–116]. This may suggest that during the compression of the denser fluid, the system may perform transitions among nearby locally stable configurations, while remaining firmly localised within a well-defined metabasin [64, 117–119].

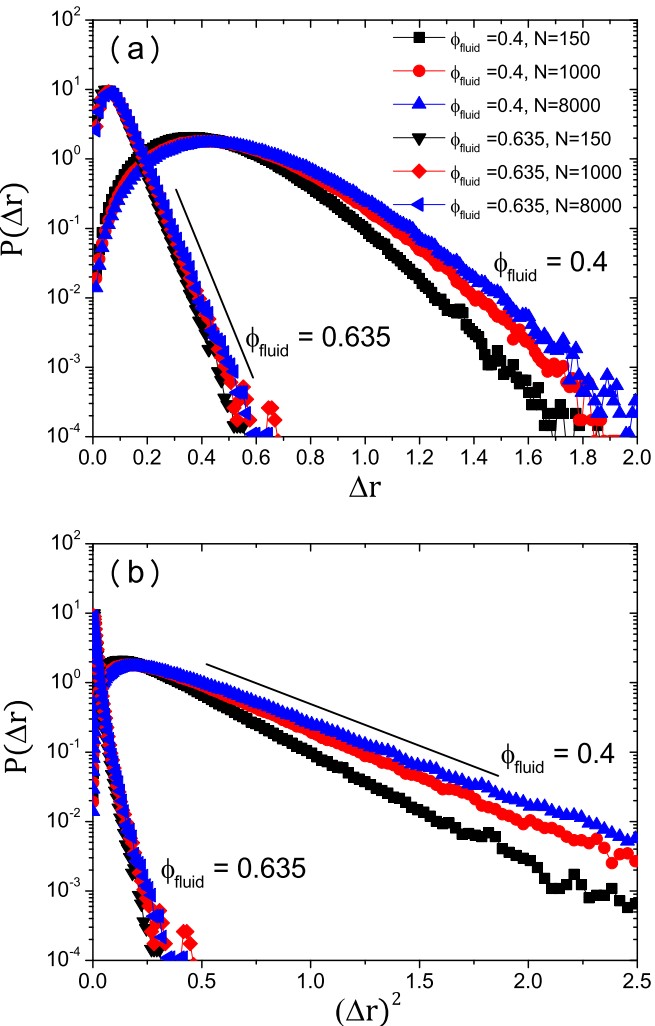

Figure 11: (a): Probability distribution function $P(\Delta r)$ of particle displacements during a compression, $\Delta r = |\mathbf{r}_i^{\text{jam}} - \mathbf{r}_i^{\text{fluid}}|$. The straight line corresponds to $P(\Delta r) \propto \exp[-a\Delta r]$, where $a$ is a constant. (b): The same distributions $P(\Delta r)$ as a function of the quantity $(\Delta r)^2$. The straight line corresponds to $P(\Delta r) \propto \exp[-b(\Delta r)^2]$, where $b$ is a constant.

## 7  Discussion and conclusions

Thanks to an efficient thermalisation algorithm which allowed us to equilibrate polydisperse hard spheres up to unprecedented packing fractions [44], we have significantly extended the study of the line of critical jamming transitions, or J-line. Our results demonstrate that isostaticity and the associated critical behavior of the pair correlation function remain unchanged along the entire J-line, while some other structural properties at larger length scales evolve qualitatively. Therefore, our results disentangle the structural properties which originate from isostaticity at the contact scale from other geometric properties at larger scale that seem unrelated or insensitive to the jamming criticality.

We found that the fraction of rattlers in jammed packings increases markedly with increasing $\phi_{\text{J}}$, which is a counterintuitive result. We confirm that a proper treatment of these rattlers is essential to reveal the critical behavior at contact scale. Rattlers also affect the organization of the packings at the neighbor scale and tend to form relatively compact clusters, which might induce subtle volume fraction fluctuations in the packing. In Ref. [47], it has been argued that

the presence of rattlers may interfere with hyperuniformity. We found that deviations from hyperuniform behavior in jammed polydisperse hard spheres become more pronounced as the volume fraction increases. However, hyperuniformity cannot be simply achieved by reducing the number of rattlers through additional compressions [24]. Moreover, hyperuniform behavior disappears completely when rattlers are removed from the computation of the spectral density, or volume fraction fluctuation [22]. Our results confirm this trend, which is in stark contrast with the jamming critical behavior seen in the pair correlation function. This would be another supporting evidence that hyperuniformity and jamming criticality are unrelated. These numerical results does not exclude the possibility that an ideal, rattler-free packing obtained through a more complex optimization process provides true hyperuniformity [47]. Along this line of thought, it would be interesting to study in more detail the relation between presence of rattlers and the violation of the hyperuniformity reported here.

We found that, above the onset volume fraction, the structure of jammed packings at the local scale closely tracks the one of the parent equilibrium fluid. This contrasts with the behavior at the contact scale, which is qualitatively very different in jammed and fluid states. In particular, as the volume fraction increases, both equilibrium fluid and jammed packings display a smooth but progressive local ordering, which we attributed to the emergence of distorted icosahedral structures. We attributed their irregular shape to the significant size dispersity of the system. We speculate that only the most regular structures, identified through the asphericity parameter used in this work or more sophisticated metrics [19], may actually provide locally stable arrangements in the dense equilibrium fluid. These ideas might find useful applications in the context of glass structure studies [46] and of investigations of the structure-dynamics relationship [90].

In a recent numerical study of polydisperse hard spheres [79], we demonstrated the growth of non-trivial static "point-to-set" correlations, which are at the core of thermodynamic pictures of the glass transition [120]. It has been suggested that such "amorphous order" could be identified with hyperuniformity [99, 102], although this link has been questioned [25]. Our results cast further doubts on this argument: point-to-set correlations grow as the volume fraction increases, while hyperuniform behavior is suppressed. The growth of amorphous order in the equilibrium fluid appears thus to compete with, rather than to enhance, hyperuniformity.

The significant suppression of finite size effects at higher $\phi_J$ suggests that the length scale causing this finite-size effect might not be related to the isostatic nature of the jamming transition, since all studied configurations are similarly isostatic. This conclusion seems to contrast with other diverging length scales near jamming [25, 100, 104, 107–109]. We notice that the finite-size effect for inherent structure energies in Lennard-Jones supercooled liquids is also suppressed when considering lower temperatures for the parent fluid [121], which is reminiscent of our observations for denser $\phi_{fluid}$ in hard spheres. Finite size effects for jammed and inherent structures may have a common physical origin, namely the different topography of the energy landscape probed in dilute or dense fluids [28, 122]. Dense fluids jam in a configuration that belongs to well-defined metabasins, whereas dilute fluids may explore a larger portion of the potential energy landscape before jamming, larger systems exploring a larger region of phase space.

We have used a system with continuous polydispersity to enhance thermalisation. However, we expect the conclusions drawn from the present model to hold more generally. Systems with discrete polydispersity, *i.e.* mixtures, are more easily amenable to theoretical investigations [101, 123–125]. Recently, it has been proposed that thermal glassy systems with continuous polydispersity can be mapped into an effective multi-component system to characterize their thermodynamic behavior [126]. Whether such an effective description is also applicable to the structural characterization of jammed states is an interesting open issue, which will likely require a careful treatment of the rattlers [124, 125].

## Acknowledgments

We thank D. Durian, P. Charbonneau, A. Ikeda, H. Ikeda, K. Kim, A. Liu, A. Ninarello, M. Pica Ciamarra, P. Sollich, and D. Vågberg for helpful discussions. The numerical simulations were partially performed at Research Center of Computational Science (RCCS), Okazaki, Japan. We thank K. Kim for providing us with CPU time in RCCS. The research leading to these results has received funding from the European Research Council under the European Unions Seventh Framework Programme (No. FP7/2007-2013)/ERC Grant Agreement No.306845. This work was supported by a grant from the Simons Foundation (No. 454933, Ludovic Berthier).

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
