# Peer review of "Exploring the jamming transition over a wide range of critical densities"

_SciPost Physics, doi:SciPost Phys. 3, 027 (2017)_

## Round 1 · Referee Report · Anonymous · 2017-7-10

Strengths

See Report

Weaknesses

See Report

Report

Great, concise, detailed and well-written and carried out study.
This paper surely should be published. I have mostly semantic and cosmetical
comments and questions - maybe some small misunderstandings. The only
three points (marked by *) where a little more discussion or even a bit more
work is needed are given below. Howver, even without much additional work
this paper can go to print after the following points are addressed in the new
version.

(Comments are written in chronological order of sequentially reading the paper
from beginning to end. If one or the other statement is falsified later, it still indicates
a possibility to improve by clarifying at the point where my question came up.)

Abstract
in the sentence “Both …” the use of critical behavior and critical densities is slightly confusing.
Does not one imply the other in different sequence? Pls rephrase. Or is it so clear that I dont
see it? The two times ‘critical’ is maybe too much.

Intro
the contact scale $k->\infty$ is maybe too much? Those short wavelengths, are they really existing
or do you just want to talk about a (decoupled) length-scale that is much smaller than particles.
Waves - mechanical ones - relate to momentum transpot on the length-scale of (2) particles still.
I thought. For radiation and fluid motion through particles, the gaps could be really active and
relevant while for heat'electric transport the contact deformation is, but not for mechanical/momentum
and mechanical energy transport. But maybe I am wrong. So drop the $\infty$, which is too strict
anyway, or better specify which waves are really at this short lenght-scale.

at ‘global scale’ k->0 it is not only fluctuations but moreover harmonic motions (low standard eigenmodes),
and it is always a finite system, so 'global’ is maybe too strong a word.

density fluctuations - is it only that?
I rather think what is relevant are the fluctuations of the density of the contact network (fabric
or just coordination number), which does exclude rattlers, since the latter entities (as well as “soft”
regions that are enclosed by stable loops) are important - especially at jamming.
Density itself displays notoriously little variation around the mean (if rattlers are included).

hard spheres configurations -> hard sphere configurations

keeping cryst. under control? - you mean largely avoiding larger crstalline regions?
And where does a local tetrahedron end and where does a crystal start?

* minute changes … I thought the protocol in [40] (overcompression) would not take very long to
generate a considerable J-range? and does not need orders of magnitude variation in time-scale
to generate some finite variation? (where the rather small range presented is due to absence
of friction). However, only later in the paper, it becomes clear that here it is also about frictionless
systems (state that earlier! e.g. in 1st line of abstract or even in the title?) so that indeed the
present system goes beyond the range in [40] which, however, was not exploring enormous
overcompression above volume-fraction 1 [which was done before, up to 10, but I dont find
back the reference]. Eventually the question remains, whether the presented results are not
due to the presently used procedure and how general that is. (I dont expect the authors to
have the answer, but the question should be raised at the end of the paper).
When I extrapolate Eq.(5) in Ref.[40] up to vol.frac.=10, I also get very close to 0.7,
while the polydispersities are somewhat different in both cases, the lower end of the J-line
almost matches. Too bad that the authors in [40] have not done this, and the present authors
use a different method … however, I think it would not be a big deal for the present authors
to do an enormous over compression and return back below jamming to see where
the jamming point has moved. Notably the present work and the preparation (over-
compression) in [40] are both isotropic. So my question is whether the both methods
give the same structures of high-density jammed systems at the end.

in the procedure-section:
why are the displacements taken from a cube - I would rather take them from a sphere to
keep the displacements isotropic - which a cube might disturb?!

the fluid branc is explored up to 0.655, which sounds like contradictory to the wide
range up to 0.7 volume fraction that is mentioned in the previous intro section.

What I miss in the description II.B. is: how close are the fluid states to jamming (how big
are the gaps?)

In the caption of Fig.1 it says: ‘rapid compression’ but in II.C a up-down-method is
explained - which is true? Also it is not clear if the left point is the Poisson phi_J,
or if the dashed line? for which size?

* In Fig.3, while the gamma=0.5 with different offset/factor is well supported by the data,
I do not see a good support of the gamma=0.41269 - and surely not for more than 1 or
maximally 2 digits. Thus the universality argument I would buy for the 0.5, but surely
NOT for the other value (0.41269) It seems to hold indeed for the lower end, but not
for the upper end of the J-line :-(

Does the present structural order have something to do with the ‘fluid with solid
features’ and ‘solid with fluid features’ speculated about in [41]? Or is it about
both states having similar features? I am not sure that I understand.

What I miss in and around V.C is the analysis of the rattler sizes.
Are they take from the small fraction or randomly from all sizes of the distribution.
Furthermore a few snapshots of the rattler-clusters might be instructive?

And also, what is the volume fraction of rattlers, where a change in rattlers
volume fraction could indicate a change in contributions from different size
particles?

An hypothesis -> A hypothesis

* Discussion of the dynamic density fluctuations and the existence of rattler clusters
(related to the statement about density-fluctuations above) could profit from some
more in-depth study. Rattlers are (do they?) NOT participating in the dynamic
(harmonic?) oscillations of the mechanically stable network and thus could indeed
be the origin of the dens-fluct.?! I agree that rattlers are not the central origin,
but with them comes the question about their clustering and their sizes.
If their sizes change, the remaining stable network has a de-facto different
size-distribution. And with that it will have different low-eigenmodes and
eigen-mode-shapes?! This might be more related to the change in \xi

Fig.10: are the data with or without rattlers? Maybe I overlooked.

Concomitantly was a new word for me - is not there a better way to
express this?

The last paragraph in VI was highly speculative and incomplete. I am not
sure that it is a good ending of the otherwise very concise study. Either
I did not understand it, so pls. rephrase, or move it to conclusion and discussion.

Conclusions:
Rattler-free packings could be obtained by growing ONLY the rattlers and NOT
by compressing the whole packing.

The icosahedra are distorted due to the size-distribution. True? then say this
clearly also in the conclusion so that it becomes more stand-alone.

The second-last sentence, could be a little more relating to the prediction in
121 and 122, where the equivalent packings to polydisperse packings were given
explicitly and - close to jamming - the importance of the rattlers contribution was
highlighted. Explicitly, the behavior of a multidisperse system and its equivalent
tridisperse system was found to be similar (under isotropic compression - shear
was not tested to my knowledge) if not identical - but ONLY after the rattlers were
removed from the size-distribution.

Last-but-not-least:

The authors might want to address more clearly in their conclusions the following issues,
either as an opinion or as a speculation (this is to the authors - I am curious about their
opinions on this.

Friction - does it matter or does it just move the J-line and its range. Does it add something
really new, or would it allow to explore jammed systems down to the transition density of ~0.53?

J-line finite size - make a more clear statement that the J-line is NOT a finite size effect
and that the finite size fluctuations have nothing to do with the protocol dependence
but rather smear it out and maybe hide it.

Shear-jamming: what would happen under shear?

Requested changes

See Report

  • validity: high
  • significance: high
  • originality: high
  • clarity: high
  • formatting: excellent
  • grammar: excellent

Author:  Daniele Coslovich  on 2017-09-13  [id 168]

(in reply to Report 1 on 2017-07-10)

We thank the referee for his/her detailed and constructive report. In our reply, we highlight the referee's comments using "email reply" style. We attach to this reply a pdf document containing four additional figures (see our reply below). A latex-diffed manuscript, which highlights all the changes made in this revision, is also available at this link https://cloud.coulomb.univ-montp2.fr/index.php/s/kyZpmL4AkEexgVV

Abstract: in the sentence “Both …” the use of critical behavior and critical densities is slightly confusing. Does not one imply the other in different sequence? Pls rephrase. Or is it so clear that I dont see it? The two times ‘critical’ is maybe too much.

We followed the referee's advise and used jamming densities'' instead ofcritical densities'' to avoid this repetition.

Intro: the contact scale $k \to \infty$ is maybe too much? Those short wavelengths, are they really existing or do you just want to talk about a (decoupled) length-scale that is much smaller than particles. Waves - mechanical ones - relate to momentum transpot on the length-scale of (2) particles still. I thought. For radiation and fluid motion through particles, the gaps could be really active and relevant while for heat'electric transport the contact deformation is, but not for mechanical/momentum and mechanical energy transport. But maybe I am wrong. So drop the $\infty$, which is too strict anyway, or better specify which waves are really at this short lenght-scale.

We agree with the referee that our original formulation could be misleading. The length scale considered here is the distance between two particles relative to contact, let's say $\delta$, (gap below $\phi_{\rm J}$, overlap above $\phi_{\rm J}$). Perfect contact corresponds to $\delta \to 0$. To avoid any ambiguity, we have used $k \sim 1/\delta \to \infty$ in the revised version of the manuscript.

at ‘global scale’ $k\to 0$ it is not only fluctuations but moreover harmonic motions (low standard eigenmodes), and it is always a finite system, so 'global’ is maybe too strong a word.

We followed the referee’s suggestion and changed global scale'' tolarge scale''.

density fluctuations - is it only that? I rather think what is relevant are the fluctuations of the density of the contact network (fabric or just coordination number), which does exclude rattlers, since the latter entities (as well as “soft” regions that are enclosed by stable loops) are important - especially at jamming. Density itself displays notoriously little variation around the mean (if rattlers are included).

As the referee points out, the geometrical features of the contact network (excluding rattlers) and its vibrational properties are important aspects of the jamming physics. However, at large scales ($k\to 0$) density or free volume fluctuations may display as well non-trivial features (e.g. hyperuniformity). In the present study, our focus is rather on the latter aspects than on the former.

hard spheres configurations $\to$ hard sphere configurations

We fixed this typo.

keeping cryst. under control? - you mean largely avoiding larger crstalline regions? And where does a local tetrahedron end and where does a crystal start?

We meant that we study only amorphous configurations for both fluid and jammed states and exclude crystallized configurations. The choice of the diameter distribution and the polydispersity of our model are optimized to be very robust against crystallization and fractionation (i.e. demixing). We rarely encountered crystallization during swap MC simulations, but when it occurs it does so in a sharp and irreversible fashion. This can be detected, for instance, via the modified bond-orientational order parameters described in Sec. V. Thus, it is easy to identify problematic samples and remove them.

  • minute changes … I thought the protocol in [40] (overcompression) would not take very long to generate a considerable J-range? and does not need orders of magnitude variation in time-scale to generate some finite variation? (where the rather small range presented is due to absence of friction). However, only later in the paper, it becomes clear that here it is also about frictionless systems (state that earlier! e.g. in 1st line of abstract or even in the title?) so that indeed the present system goes beyond the range in [40] which, however, was not exploring enormous overcompression above volume-fraction 1 [which was done before, up to 10, but I dont find back the reference]. Eventually the question remains, whether the presented results are not due to the presently used procedure and how general that is. (I dont expect the authors to have the answer, but the question should be raised at the end of the paper). When I extrapolate Eq.(5) in Ref.[40] up to vol.frac.=10, I also get very close to 0.7, while the polydispersities are somewhat different in both cases, the lower end of the J-line almost matches. Too bad that the authors in [40] have not done this, and the present authors use a different method … however, I think it would not be a big deal for the present authors to do an enormous over compression and return back below jamming to see where the jamming point has moved. Notably the present work and the preparation (over- compression) in [40] are both isotropic. So my question is whether the both methods give the same structures of high-density jammed systems at the end.

First of all, we have added the word “frictionless” in the first line of abstract and the introduction part to emphasize absence of the friction in our study.

We, however, respectfully disagree with the rest of the comment; it is very difficult to increase dramatically the range of jamming densities.

We think that achieving isostatic jammed packings as dense as those of this study is a really tough optimization process, requiring long simulation time scales to properly organize dense amorphous configurations. This is because, conceptually, high density jammed packings are located very deep in the energy (volume) landscape. Compression of well-thermalized high density hard sphere fluids is a straightforward protocol to get high density jammed packings, but it requires enormous simulation time scale for thermalization of high density fluids. The swap Monte Carlo setting employed in this study shortcuts this tough thermalization process. This simulation ``trick'' enables us to access very well-thermalized fluid states within a short simulation time scale.

It is important to realize that other jamming protocols such as the one in [Kumar and Luding, Granular Matter, 18, 1 (2016)] could in principle attain very high density $\phi_{\rm J}$ as long as the protocol utilizes successive optimization process to penetrate deep in the landscape (e.g. repeated over-compressions). However, we would expect that, in the absence of some simulation ``trick'', the number of cycles (or $M$ in the above paper) and corresponding simulation time scales required to reach similar value of $\phi_{\rm J}$ as our study ($\phi_{\rm J} \sim 0.7$) would be very large.

We think that both protocols (ours and over-compressions) would give the same structural properties of high density jammed systems at the contact scale, as long as the packings are isostatic. Also, both protocols would reproduce the similar qualitative trends, such as the substantial increase of icosahedral structures. However, quantitative features such the fraction of rattlers would depend on the protocol. Thus, the absolute values of the local order parameters and the deviation from hyperuniformity would be dependent on protocols.

in the procedure-section: why are the displacements taken from a cube - I would rather take them from a sphere to keep the displacements isotropic - which a cube might disturb?!

Equilibrium properties measured in a Monte Carlo simulation do not depend on specific shape of region (either cube or sphere) used to draw random displacements. Thus, we follow conventional shape of the region (cube), see e.g. [Allen and Tildesley, "Computer simulation of liquids"]. But a sphere would do just as well!

the fluid branch is explored up to 0.655, which sounds like contradictory to the wide range up to 0.7 volume fraction that is mentioned in the previous intro section.

What I miss in the description II.B. is: how close are the fluid states to jamming (how big are the gaps?)

In this study, we obtain a wide range of the jamming transition volume fractions, $\phi_{\rm J} \simeq 0.65 - 0.7$, by compressing a range of equilibrium hard sphere fluids prepared at $\phi_{\rm fluid}=0.4-0.655$. To distinguish between $\phi_{\rm J}$ and $\phi_{\rm fluid}$, we added the subscript ``fluid'' in the sentence mentioned by the referee.

Moreover, to indicate the relation between the fluid and jammed states more clearly, we now include the equation of state in Fig. 1 (a). The equilibrium fluid state and jammed state are located at finite reduced pressures $p$ and at $p \to \infty$, respectively.

In the caption of Fig.1 it says: ‘rapid compression’ but in II.C a up-down-method is explained - which is true? Also it is not clear if the left point is the Poisson $\phi_{\rm J}$, or if the dashed line? for which size?

We only use the algorithm described in II. C throughout the paper. This algorithm amounts to a rapid non-equilibrium compression in the sense that the hard spheres fluids are compressed up to the jammed packing. However, under the hoods, the algorithm performs tiny compressions and decompressions (up-down) in order to find precise isostatic packing for a given configuration. To avoid confusion, we have dropped "rapid" from the sentence mentioned by the referee.

Also, the horizontal dashed line in Fig.1(b) is $\phi_{\rm J}$ starting from a Poisson distributed configuration with $\phi_{\rm fluid}=0.3$ for $N=8000$. We have clarified this in the revised manuscript.

  • In Fig.3, while the $\gamma=0.5$ with different offset/factor is well supported by the data, I do not see a good support of the $\gamma=0.41269$ - and surely not for more than 1 or maximally 2 digits. Thus the universality argument I would buy for the 0.5, but surely NOT for the other value (0.41269) It seems to hold indeed for the lower end, but not for the upper end of the J-line :-(

We do not have a firm explanation concerning the range of the critical regime for the critical power law behavior in $g(x)$. One possible reason why the critical regime seems to shrink with increasing $\phi_{\rm J}$ is that, according to the phase diagram drawn in [Charbonneau et al., Annu. Rev. Condens. Matter Phys. 8, 265 (2017)], the marginal glass phase associated with the jamming criticality is getting narrower with increasing $\phi_{\rm J}$ and the stable glass phase approaches the marginal glass phase, possibly interfering with the critical behavior.

To emphasize the observed critical regime for $\gamma=0.41269$, we have added a gray shaded region in Fig. 3.

Does the present structural order have something to do with the ‘fluid with solid features’ and ‘solid with fluid features’ speculated about in [41]? Or is it about both states having similar features? I am not sure that I understand.

Our simulations demonstrate that the fluid and jammed states share similar local structures at higher densities. In both states, distorted icosahedra become more abundant with increasing the density and are expected to provide locally stable, "solid-like" behavior. In this sense, yes, we think our findings are related to the ideas of [Luding, Nat. Phys. 12, 531 (2016)].

Indeed, the idea of a "fluid with solid features" lies behind several recent works on glassy dynamics, see [C.P. Royall and S. R. Williams, Phys. Rep. 560, 1 (2015)] for a review. To quantitatively investigate this idea for the present system, one could study structure-dynamics relation as done in [Hocky et al. Phys. Rev. Lett. 113, 157801 (2014)]. Similarly, the intriguing picture of a "solid with fluid features" may be pertinent for the description of shear flow. By observing thermal motion or the mechanical response of the particles participating in distorted icosahedra, one could identify a connection between local structure and plastic events. For a recent work along these lines see e.g. [R. Pinney, T.B. Liverpool, and C.P. Royall, J. Chem. Phys. 145, 234501 (2016)].

What I miss in and around V.C is the analysis of the rattler sizes. Are they take from the small fraction or randomly from all sizes of the distribution. Furthermore a few snapshots of the rattler-clusters might be instructive?

And also, what is the volume fraction of rattlers, where a change in rattlers volume fraction could indicate a change in contributions from different size particles?

Following the referee's advise, we have analyzed the distribution of the rattler particle diameter, see Fig. 6(a) of the revised manuscript. We find that the rattlers tend to be the smallest particles. Interestingly, the shape of the distribution does not change appreciably along the J-line. We have also added snapshots showing the rattlers spatial distribution at two different $\phi_{\rm J}$'s as suggested by the referee.

We think the referee refers to the "average" volume fraction of the rattlers. One practical way to evaluate this quantity is by measuring the volume of the Voronoi cells around rattlers. We find that the averaged Voronoi cell volume of rattlers scales like the overall volume fraction $phi_{\rm J}$ along the J-line. This trivial effect confirms our conclusion that there is no important change in the composition of the rattlers along the J-line.

An hypothesis $\to$ A hypothesis

We fixed this typo.

  • Discussion of the dynamic density fluctuations and the existence of rattler clusters (related to the statement about density-fluctuations above) could profit from some more in-depth study. Rattlers are (do they?) NOT participating in the dynamic (harmonic?) oscillations of the mechanically stable network and thus could indeed be the origin of the dens-fluct.?! I agree that rattlers are not the central origin, but with them comes the question about their clustering and their sizes. If their sizes change, the remaining stable network has a de-facto different size-distribution. And with that it will have different low-eigenmodes and eigen-mode-shapes?! This might be more related to the change in $\xi$

See Fig. R1 in the file attached to this reply.

The referee argues that low-frequency normal modes (so-called soft modes) and their shape might change along the J-line. Prompted by this remark, we performed a normal mode analysis of the contact network (without rattlers) extracted from our jammed packings and computed the density of states $D(\omega)$ and the participation ratio. We found that although the soft modes are slightly more abundant at higher $\phi_{\rm J}$, the overall shape of $D(\omega)$ remains essentially the same, see Fig. R1(a). We also found that the participation ratio shows similar scatter plots for higher and lower $\phi_{\rm J}$'s, suggesting that the structure of normal modes does not change along the J-line, see Fig. R1(b).

Since in this work we are mostly concerned with the static and structural properties of jammed packings at high densities, we do not wish to include these figures in the main text. They will nonetheless remain available as part of this submission, should the paper be accepted in SciPost.

Fig.10: are the data with or without rattlers? Maybe I overlooked.

$P(\Delta r)$ is computed for all particles including rattlers. We have added one sentence to explain this in the text.

Concomitantly was a new word for me - is not there a better way to express this?

We have changed concomitantly'' tosimultaneously''.

The last paragraph in VI was highly speculative and incomplete. I am not sure that it is a good ending of the otherwise very concise study. Either I did not understand it, so pls. rephrase, or move it to conclusion and discussion.

We have moved this paragraph to the discussion and conclusions section.

Conclusions: Rattler-free packings could be obtained by growing ONLY the rattlers and NOT by compressing the whole packing.

See Fig. R2 in the file attached to this reply.

Following the referee's advice, we tried to achieve rattler-free isostatic packings by inflating and decompressing only the rattlers. The idea is the following. First, only the diameters of rattlers are inflated so that the fraction of rattlers becomes zero, $N_{\rm r}/N=0$, taking the averaged contact number $Z>6$ (hyperstatic). Then, the diameters of rattlers are shrinked with successive energy minimization toward hard sphere packing. In Fig. R2, we show the volume fraction dependence of $Z$ and of $N_{\rm r}/N$ for this "unjamming" process of rattlers. We find that the system goes back to the original $\phi_{\rm J}$ with the same $Z$ and $N_{\rm r}/N$, suggesting that the positions of rattlers are determined by the surrounding non-rattler particles (or contact network). We also tried to optimize all particles of the system after inflating the rattlers only. In this case, the system becomes hypostatic ($Z<6$) and even $N_{\rm r}/N$ takes larger than original value (not shown).

From these observations, we tentatively conclude that the positions of the rattlers are much more optimized than we expected, and inflating the rattlers does not produce isostatic, rattler-free packings.

In short, getting rid of rattlers appears as a difficult problem, that no simple-minded solution is able to solve.

The icosahedra are distorted due to the size-distribution. True? then say this clearly also in the conclusion so that it becomes more stand-alone.

Yes, we mentioned already in Sec. V B that we attribute the presence of distorted, irregular icosahedra to the size polydispersity of the system. We have added a similar comment in the conclusions of the revised manuscript.

The second-last sentence, could be a little more relating to the prediction in 121 and 122, where the equivalent packings to polydisperse packings were given explicitly and - close to jamming - the importance of the rattlers contribution was highlighted. Explicitly, the behavior of a multidisperse system and its equivalent tridisperse system was found to be similar (under isotropic compression - shear was not tested to my knowledge) if not identical - but ONLY after the rattlers were removed from the size-distribution.

We second the referee's remark and we now mention in the main text that a careful treatment of rattlers is important for an effective mapping between multi-component and continuous polydisperse jammed packings.

Friction - does it matter or does it just move the J-line and its range. Does it add something really new, or would it allow to explore jammed systems down to the transition density of ~0.53?

We do not have a firm opinion on the role of friction. This is of course an interesting open question for a future investigation.

J-line finite size - make a more clear statement that the J-line is NOT a finite size effect and that the finite size fluctuations have nothing to do with the protocol dependence but rather smear it out and maybe hide it.

We now mention in the main text that our data conclusively demonstrate that the J-line is not due to a finite size effect and that it remains well-defined in the thermodynamic limit.

Shear-jamming: what would happen under shear?

Our simulation setting, which combines polydispersity and swap Monte Carlo, yields very deeply equilibrated supercooled fluids. Compressing or quenching such extremely viscous fluids further enables us to produce solids that lie very deep in the landscape, as demonstrated by the very high $\phi_{\rm J}$ packings. Currently we are interested in how such ultrastable amorphous solids break or yield when shear strain is imposed. Shear yielding of ultrastable amorphous solids would show qualitatively different behavior than ordinary solids.

Another interesting subject would be the relation between shear yielding and shear jamming. To the best of our knowledge, there is no clear understanding of which condition the system yield or jam, or what is the interplay between yielding and jamming. A non-trivial phase diagram of shear yielding and jamming has been recently proposed by [Urbani and Zamponi, Phys. Rev. Lett. 118, 038001 (2017)]. We think this might lead to an interesting research avenue.

Attachment:

supplement.pdf

---

## Round 1 · Referee Report · Anonymous · 2017-7-11

Strengths

Overall, it is very interesting to see that the width of the J-line can be widened, using tailored distributions, with all the critical behaviour remaining intact. The attempt to characterise the structural properties of these jammed states at different scales is also worthwhile. Therefore, I think this is a useful contribution that should be eventually published.

Weaknesses

None

Report

1) The work is done for a specific distribution of particle sizes, with a specific width (polydispersity 23%). I would guess that this choice of the distribution helps in pushing the critical densities to higher values. Is that the case ? It would be illustrative to show the results for a few more polydispersities, may be at least one larger and one smaller to see how the range of the J-line is dependent on this, if at all. Some earlier work, from Wilding, Sollich and co-workers, have suggested that large polydispersities would result in fractionation at large densities. The authors should comment on this vis-a-vis the jammed states.

2) Section B is very sketchy. For completeness, the authors should plot the distribution of sizes and also demonstrate that they are sitting on the equilibrium line. Otherwise, readers have to wade through other papers to follow this. In fact, in that section there are not
sufficient references for the readers, even to guide them through it.

4) The authors themselves point out that the number of rattlers are fairly large in the most dense system. They also try to discuss the spatial organisation of these rattlers. For this some snapshots might help to get an idea of how the rattlers are spatially distributed for
the different J points achieved. Is there some pattern regarding which sizes, large or small, are becoming rattlers ?

5) The system sizes studied are not optimum to discuss about the low k behaviour in chi(k) and related signatures of or deviations from
hyperuniformity (Fig.7d). Please discuss/comment

6) The discussion on finite size effects should perhaps come earlier, probably after III. Better data is needed for phi_fluid=0.635, for
large N.

Requested changes

Each of the 5 points in the Report above requests some clarifications, additions and changes.

  • validity: high
  • significance: high
  • originality: high
  • clarity: good
  • formatting: good
  • grammar: good

Author:  Daniele Coslovich  on 2017-09-13  [id 169]

(in reply to Report 2 on 2017-07-11)

We thank the referee for taking the time to contribute with this report and for his/her constructive criticisms. In our reply, we highlight the referee's comments using "email reply" style. A latex-diffed manuscript, which highlights all the changes made in this revision, is also available at this link https://cloud.coulomb.univ-montp2.fr/index.php/s/kyZpmL4AkEexgVV

The work is done for a specific distribution of particle sizes, with a specific width (polydispersity 23\%). I would guess that this choice of the distribution helps in pushing the critical densities to higher values. Is that the case ? It would be illustrative to show the results for a few more polydispersities, may be at least one larger and one smaller to see how the range of the J-line is dependent on this, if at all. Some earlier work, from Wilding, Sollich and co-workers, have suggested that large polydispersities would result in fractionation at large densities. The authors should comment on this vis-a-vis the jammed states.

Swap Monte Carlo (MC) is a very efficient method to explore the phase space of polydisperse fluid systems. Previous simulation studies using this method were hampered by crystallization or fractionation (demixing) at low temperature or high density. In particular, as the referee points out, large polydispersities lead to fractionation. On the other hand, if the polydispersity is too small, the system will quickly crystallize once brought in the metastable regime. In [Ninarello, Berthier, and Coslovich, Phys. Rev. X, 7, 021039 (2017)] we varied several model parameters (including e.g. softness and non-additivity) of polydisperse particles and studied the robustness against crystallization and fractionation. The specific model parameters we use here allows one to enjoy all the benefits of the swap MC without having to deal with the above-mentioned instabilities. With larger or smaller polydispersities, it might be difficult to stretch the J-line as much as we do here. We note, however, that it may be possible to produce even more stable packings using a non-additive variant of the model, as our recent work published in Phys. Rev. X suggests.

In other words, other polydispersities would produce a smaller range of J-line, because thermalisation with no ordering would be more difficult.

We have mentioned these aspects at the end of Section II A and we have cited the works by Wilding and Sollich on fractionation of polydisperse spheres (new Refs. [50, 51]).

Section B is very sketchy. For completeness, the authors should plot the distribution of sizes and also demonstrate that they are sitting on the equilibrium line. Otherwise, readers have to wade through other papers to follow this. In fact, in that section there are not sufficient references for the readers, even to guide them through it.

Following the referee's suggestion, we have added the diameter distribution employed in this study in Fig. 6(a). Also, in Fig. 1(a), we included the equation of state to highlight the difference between the equilibrium fluid state and jammed states in the phase diagram.

As requested, we have added appropriate references in Sec. B (Refs. 44, 48, 50, 53-59 in the revised manuscript).

The authors themselves point out that the number of rattlers are fairly large in the most dense system. They also try to discuss the spatial organisation of these rattlers. For this some snapshots might help to get an idea of how the rattlers are spatially distributed for the different J points achieved. Is there some pattern regarding which sizes, large or small, are becoming rattlers ?

We have added snapshots of rattlers in Fig. 7 to give an idea about their real space organization. The rattlers are distributed rather homogeneously in this representation. Also, we analyzed the diameter distribution for rattlers and found that rattlers tend to be smaller particles. This is now shown in Fig. 6 (a).

The system sizes studied are not optimum to discuss about the low k behaviour in $\chi(k)$ and related signatures of or deviations from hyperuniformity (Fig.7d). Please discuss/comment

As the referee points out, $N=8000$ may not be big enough to conclusively demonstrate the existence or absence of hyperuniformity. However, it has been shown that a signature of hyperuniformity, $\chi(k) \propto k$, is already observed in smaller system as a precursor'' of hyperuniformity (see, e.g., [Atkinson et al., Phys. Rev. E, 94, 012902 (2016)]). Indeed we observe such linear behavior at low $\phi_{\rm J}$ in $N=8000$ systems. Our main conclusion about hyperuniformity is that suchprecursors'' are systematically destroyed with increasing $\phi_{\rm J}$, suggesting that hyperuniformity would not be retained in much bigger systems or at smaller $k$. Therefore, we conclude that very large systems are needed to decide about hyperuniformity when those systems are very close to being hyperuniform; in our case, these deviations are so strong that the system sizes we study are good enough to support our claims.

The discussion on finite size effects should perhaps come earlier, probably after III. Better data is needed for $\phi_{\rm fluid}=0.635$, for large N.

Our paper is structured so as to analyze structural properties at progressively larger length scales, ranging from the contact scale to large scales. We think that the finite size scaling analysis contains information about large length scales associated to the jamming transition. Therefore, we prefer to keep the corresponding discussion at the end of Sec. VI and leave the flow of the presentation as in the original version of the manuscript.

To comply with the referee's remark, we have increased the statistics twice for phi_fluid=0.635, for N>=2000. This leads to cleaner data in Fig. 10(b).

---

## Round 2 · Author Response

Dear Editor,

We thank the referees for their insightful and constructive reports. As detailed in our replies to the referees, we have addressed all their remarks and revised our manuscript accordingly. We hope that we these changes the manuscript will be considered suitable for publication in SciPost.

Sincerely,

Daniele Coslovich

---

## Round 2 · List of Changes

- Sec. II A: add details on the choice of the model parameters; add appropriate references
- Add equation of state as Fig. 1(a)
- Add shaded region in Fig. 3 to highlight the regime where critical scaling holds
- Add diameter distribution for rattlers in Fig. 6(a)
- Add snapshots depicting the spatial organization of rattlers as Fig. 7
- Improve statistics in the analysis of finite size effects; update Fig. 10 and estimates of ϕJ and ν accordingly.
- Rename section VII "Discussion and conclusions"
- Move the final paragraph of Sec. VI to Sec. VII
- Minor changes: reword and clarify some sentences (see replies for details), fix typos

---

## Editorial Decision

published